# Composition and cycling of dissolved organic matter from tropical peatlands of coastal Sarawak, Borneo, revealed by fluorescence spectroscopy and PARAFAC analysis

Yongli Zhou[1], Patrick Martin[1], Moritz Müller[2]

[1] Asian School of the Environment, Nanyang Technological University, Singapore 639798 Singapore

[2] Swinburne University of Technology, Faculty of Engineering, Computing and Science, 93350 Kuching, Sarawak, Malaysia

Correspondence to: Yongli Zhou (zhou0303@e.ntu.edu.sg)

**Abstract.** Southeast Asian peatlands supply ~10 % of the global flux of dissolved organic carbon (DOC) from land to the ocean, but the biogeochemical cycling of this peat-derived DOC in coastal environments is still poorly understood. Here, we use fluorescence spectroscopy and parallel factor (PARAFAC) analysis to distinguish different fractions of dissolved organic matter (DOM) in peat-draining rivers, estuaries, and coastal waters of Sarawak, Borneo. The terrigenous fractions showed high concentrations at freshwater stations within the rivers, and conservative mixing with seawater across the estuaries. The

autochthonous DOM fraction, in contrast, showed low concentrations throughout our study area at all salinities. The DOM pool was also characterized by a high degree of humification in all rivers and estuaries up to salinity 25. These results indicate a predominantly terrestrial origin of the riverine DOM pool. Only at salinities >25 did we observe an increase in the proportion of autochthonous relative to terrestrial DOM. Natural sunlight exposure experiments with river water and seawater showed high photolability of the terrigenous DOM fractions, suggesting that photodegradation may account for the observed changes

in DOM composition in coastal waters. Nevertheless, we estimate based on our fluorescence data that at least 20 %–25 % of the DOC at even our most marine stations (salinity >31) was terrestrial in origin, indicating that peatlands likely play an important role in the carbon biogeochemistry of Southeast Asian shelf seas.

# 1 Introduction

Tropical peatlands store around 100 Pg of carbon, of which 55 % is found in Southeast Asia (Page et al., 2011; Dargie et al., 2017), mostly on the islands of Sumatra and Borneo (Dommain et al., 2014). The rivers draining Southeast Asia's peatlands export large quantities of terrigenous dissolved organic carbon (tDOC), accounting for ~10 % of the global land-to-ocean DOC
flux of 0.2–0.25 Pg C yr$^{-1}$ (Meybeck, 1982; Baum et al., 2007; Moore et al., 2011; Dai et al., 2012). Terrigenous dissolved organic matter (tDOM) can play significant roles in aquatic environments: tDOM is susceptible to decomposition processes that can remineralize a considerable proportion (40 %–50 %) of it in estuaries and shelf seas (Fichot and Benner, 2014; Kaiser et al., 2017). Remineralization of tDOM contributes to maintaining net heterotrophy and $CO_2$ outgassing in some inner estuaries and ocean margins (Borges et al., 2006; Cai, 2011; Chen and Borges, 2009), potentially causing significant seawater
acidification (Alling et al., 2012; Semiletov et al., 2016). tDOM remineralization can also supply inorganic nutrients (Vähätalo and Zepp, 2005; Stedmon et al., 2007; Aarnos et al., 2012).

tDOM is increasingly recognized as labile to both photodegradation (Aarnos et al., 2018; Helms et al., 2014; Hernes and Benner, 2003) and biodegradation (Moran et al., 2000; Wickland et al., 2007; Carlson and Hansell, 2014). For example, photodegradation was estimated to account for 70 %–95 % of total DOM processing in the arctic lakes and rivers (Cory et al.,
2014). In the Congo River, which drains extensive tropical peatlands, >95% of the lignin phenols and 45 % of the total DOC pool are labile to photodegradation, which thus reduces average molecular weight and aromatic structures (Spencer et al., 2009; Stubbins et al., 2010). Microbial processing can be responsible for a major carbon loss as well, but typically results in shifts of DOM optical properties in the opposite direction to those caused by photodegradation (Moran et al., 2000). Moreover, biodegradation shows a preference for hydrophilic compounds, especially amino acid-like fractions (Wickland et al., 2007;
Benner and Kaiser, 2011). On the Louisiana Shelf, the remineralization of tDOM from the Mississippi River was found to be dominated by biodegradation rather than photodegradation (Fichot and Benner, 2014). The fate of tDOM in aquatic environments also depends on the interaction between these two processes, exemplified by the elevated biodegradability of tDOM after partial photodegradation, which decomposes the bio-resistant compounds beforehand (Miller and Moran, 1997; Moran and Zepp, 1997; Moran et al., 2000; Smith and Benner, 2005).

However, our knowledge of the biogeochemical cycling of peat-derived DOM in Southeast Asia is still limited. Although several studies have shown that peatland-draining blackwater rivers in Sumatra and Borneo carry extremely high DOC concentrations (3000–5500 μmol L$^{-1}$, or 36–66 mg L$^{-1}$) with a predominantly terrestrial origin (Alkhatib et al., 2007; Baum et al., 2007; Rixen et al., 2008; Harun et al., 2015, 2016; Müller et al., 2015, 2016; Cook et al., 2017), more detailed analysis of the chemical composition of peat-derived DOM, and determination of its lability to different degradation processes, are mostly
lacking. Moreover, most of these studies did not sample beyond the upper estuaries. Notably, however, Southeast Asian peat-draining rivers have low pCO$_2$ relative to the high DOC concentrations (Müller-Dum et al., 2019; Müller et al., 2015, 2016; Wit et al., 2015), which implies that there is little biogeochemical processing of tDOM within the rivers. However, given that tDOM is increasingly recognized as potentially labile in aquatic environments, more studies are needed to characterize

Southeast Asian tDOM and its biogeochemical cycling across the full continuum from freshwater through estuaries to the coastal sea. This is particularly urgent in light of the extensive land-use changes in Southeast Asia over the past three decades, especially the conversion of peatlands to industrial plantations (Miettinen et al., 2016), which appear likely to have increased the riverine flux of tDOC (Moore et al., 2013).

5 A companion study by Martin et al. (2018) revealed high DOC and CDOM concentrations, and low CDOM spectral slopes, in peat-draining rivers in Sarawak, indicating large tDOM input, and conservative mixing of DOC with seawater. However, the composition of the organic matter and the cycling processes of different fractions after export from peatlands still remain unknown. In this study, using fluorescence spectroscopy and parallel factor (PARAFAC) analysis in the same region, we aimed to: (1) further resolve the chemical composition of DOM and the biogeochemical fate of individual DOM fractions during 10 riverine transport; (2) infer spatial patterns of tDOM degradation; and (3) estimate the potential contribution of photodegradation to the removal and modification of tDOM.

## 2 Methods

### 2.1 Sampling

The study region, sampling methods, and the photodegradation experiments have already been described in detail by Martin 15 et al. (2018). Briefly, we sampled six rivers (the Rajang, Sematan, Samunsam, Maludam, Sebuyau, and Simunjan rivers), their estuaries, and open coastal waters in early March, June and September 2017 (Fig. 1). These months correspond to the end of the wet northeast monsoon, during the drier southwest monsoon, and the end of the southwest monsoon, respectively. However, this equatorial climate does not have distinct wet and dry seasons: monthly rainfall is quite high year-round (200–400 mm, Sa'adi et al. (2017)). One additional sample was collected from the Lundu River estuary in September. During each expedition, 20 weather conditions on most days were part cloudy / part sunny, with heavy rain showers of a few hours' duration occurring across small spatial scales on many days. No extreme weather was encountered. All samples were collected within the upper 1 m and filtered on the same day through 0.2-µm pore-size Anodisc filters (47-mm diameter). The all-glass filtration system was cleaned with 1 M HCl and deionized water (18.2 MΩ cm$^{-1}$, referred to as "DI water" below), and filters were pre-rinsed with both DI water and sample water. Filtered samples (30 mL each) for fluorescence and absorbance spectroscopy were then 25 preserved with 150 µL of 10 g L$^{-1}$ NaN$_3$, following Tilstone et al. (2002), stored in amber borosilicate vials with PTFE-lined septa at +4° C and analyzed within 1.5 months of collection.

All six rivers drain peatlands, but to very varying degrees. The Rajang River catchment is dominated by mineral soils, and peatlands are only found within the delta, downstream of the town of Sibu (Staub et al., 2000; Gastaldo, 2010). The Sematan and Lundu rivers also drain catchments with more limited peatlands and a higher proportion of mineral soil. In contrast, the 30 Samunsam, Maludam, Sebuyau, and Simunjan rivers drain catchments that consist to a large extent of peatlands, and these four rivers are considered blackwater rivers. Mangroves are found along the estuaries of all six rivers. Following the companion study by Martin et al. (2018), we distinguish between sampling stations in the eastern region (Rajang River and coastal water

stations east of Kuching city), the western region (Sematan and Samunsam rivers, and coastal water stations west of Kuching), and the remaining three blackwater rivers.

## 2.2 Photodegradation

Photodegradation experiments were conducted in June and September by filtering (0.2 µm, Anodisc) water samples from the Rajang River, the Samunsam River and eastern region seawater, and exposing them to natural sunlight in 150-mL quartz bottles for 3–6 days. Filtered water for these experiments was first collected in an acid-washed, 1-L glass bottle to create a homogenous sample before being poured into the acid-washed quartz bottles. Triplicate bottles from each site were wrapped in aluminum foil and black plastic as dark controls. The bottles were repetitively subsampled every 1–3 days, and samples

were preserved as above. Martin et al. (2018) showed that all experiments received approximately equal sunlight irradiance over time, so for simplicity we present our results as a function of exposure time.

## 2.3 Absorbance and fluorescence measurement and data processing

The absorbance measurement methods are described in detail by Martin et al. (2018). Briefly, absorbance spectra were measured using a dual-beam Thermo Evolution 300 spectrophotometer with 10-cm, 1-cm, or 0.2-cm pathlength quartz cuvettes.

In March, when the 2-mm cuvette was unavailable, high-absorbance samples were diluted ten-fold with DI water and measured in a 1-cm cuvette. Laboratory reagent blanks of 30 mL DI water + 150 µL of 10 g L$^{-1}$ NaN$_3$ were measured and subtracted from all spectra, because NaN$_3$ absorbs strongly at wavelengths shorter than 300 nm (McDonald et al., 1970). Because the NaN$_3$ concentration was identical between samples with very little variation, the subtraction of this blank did not introduce large uncertainties even for low-CDOM samples, as we show in the Supplementary Information 1. Martin et al. (2018)

calculated Napierian absorption coefficients at 350 nm ($a_{350}$, a measure of CDOM concentration), the spectral slope between 275 nm and 295 nm ($S_{275-295}$, which is established as a tracer for terrestrial origin and is related to average DOM molecular weight, Helms et al. (2008)), and specific UV absorbance at 254 nm (SUVA$_{254}$, a measure of the proportion of aromatic compounds in the DOM pool, and also a tracer of tDOM, Weishaar et al. (2003)).

Fluorescence excitation-emission matrices (EEM) were measured using a Jobin Yvon Horiba Fluoromax-4 fluorometer

(excitation: 250–450 nm at 5-nm intervals; emission: 290–550 nm at 2-nm intervals; both bandwidths 5 nm). To minimize self-quenching of fluorescence, samples with high absorbance in March were diluted ten-fold with DI water. In September, samples with high absorbance were measured undiluted in a 3-mm pathlength cuvette. All other samples were measured undiluted in a 1-cm pathlength cuvette. Laboratory reagent blanks of DI water with NaN$_3$ were measured at appropriate dilution in both cuvettes for blank subtraction. NaN$_3$ did not contribute any blank fluorescence. Fluorescence signals were normalized

to the lamp reference intensity, with spectral corrections applied by the instrument software.

Data were further processed with the MATLAB drEEM toolbox (Murphy et al., 2013) to (1) correct for inner filter effects (IFE) following Kothawala et al. (2013) using the total absorbance of each sample, thus accounting for the presence of NaN$_3$

in each sample, (2) convert fluorescence intensities to Raman Units (R.U.) based on the area of the water Raman peak (Lawaetz and Stedmon 2008), (3) subtract blanks, and (4) where necessary correct for sample dilution. First-order Raman scattering and second-order Rayleigh scattering were completely removed, while second-order Raman scattering and first-order Rayleigh scattering were smoothed by interpolation. Caution is needed for three samples, Samunsam station 4 and 5 (March) and Sebuyau station 4 (March), due to the $A_{total}$ (sum of absorbance at each pair of excitation and emission wavelength) exceeding 1.5 in the short wavelength region of the EEMs (Fig. S3), which resulted in invalid IFE correction (Kothawala et al., 2013) but this only potentially affected the PARAFAC results of C5 and C3 (see below) for these three samples. The IFE correction is fully valid for all other samples.

Because our EEM data were corrected using instrument-specific correction factors, we calculated the fluorescence index, FI, as the ratio of emission intensity at 470 nm to that at 520 nm, at excitation 370 nm, following Cory et al. (2010) (Eq. 1):

$$FI = \frac{Ex370, Em470}{Ex370, Em520} \quad (1)$$

We also calculated the humification index, HIX, following Ohno (2002) (Eq. 2):

$$HIX = \frac{Ex255, \sum Em(434 \rightarrow 480)}{Ex255, \sum Em(434 \rightarrow 480) + Ex255, \sum Em(300 \rightarrow 346)} \quad (2)$$

where $Ex255, \sum Em(x \rightarrow y)$ is the integrated area under the emission spectrum from x nm to y nm excited at 255 nm (note that Ohno (2002) originally used excitation 254 nm).

## 2.4 PARAFAC analysis

A total of 225 corrected EEMs from field samples and the photodegradation experiments were used for PARAFAC analysis using the MATLAB drEEM toolbox, which decomposes the variation between EEMs in a dataset into multiple mathematically independent components representing different organic compound classes, with different sources, biogeochemical properties and behaviors (Bro and Kiers, 2003; Stedmon, et al., 2003; Stedmon and Bro, 2008; Murphy et al., 2013). Four outliers with abnormal EEM spectra or unusually high leverage over the model were removed from this study. A five-component model was generated and validated by residual examination and split-half analysis. We compared our PARAFAC components with components from previous studies in the OpenFluor database (Murphy et al., 2014) to identify the possible source and biogeochemical properties of our components. PARAFAC components are quantified as the highest score value at the emission maxima, known as the fluorescence intensity at the maximum (Fmax), which is taken as a measure of the relative concentration of each component within a dataset (Murphy et al., 2013). We report our values in Raman Units (R.U.), which can be roughly converted to Quinine Sulfate Units (QSU) by multiplying by 48.9 (Stedmon and Markager, 2005a).

# 3 Results

## 3.1 Biogeochemical setting

A detailed discussion of the DOC and CDOM distribution and characteristics in the study region can be found in Martin et al. (2018). Briefly, high DOC concentrations (1,200–4,400 µmol L$^{-1}$), high CDOM absorption coefficients ($a_{350}$ of 50–200 m$^{-1}$),

and CDOM properties with clear terrestrial signals were found in all of the four blackwater rivers. The highest DOC was found in the Maludam River (3,100-4,400 µmol L$^{-1}$). The Rajang and Sematan rivers had lower DOC concentrations (120-450 µmol L$^{-1}$) and less CDOM ($a_{350}$ of 3–11 m$^{-1}$), consistent with a greater proportion of mineral soil rather than peat in these two catchments. DOC and CDOM in all estuaries showed mostly conservative mixing with seawater, except in the Rajang River, where additional organic matter input in the estuary was inferred from the DOC distribution. DOC at the stations furthest from

the coast was as low as 76 µmol L$^{-1}$. A predominantly terrestrial origin of DOM in the rivers was inferred from the low CDOM $S_{275-295}$ (0.0102–0.0144) and high SUVA$_{254}$ (3.08–6.89) (Martin et al., 2018).

Chlorophyll-*a* concentrations were low at all stations, mostly below 3 µg L$^{-1}$, and never exceeding 5.5 µg L$^{-1}$ (Martin et al., 2018). In the rivers, these oligotrophic conditions are most likely a result of light limitation due to high sediment (at the Rajang and Sematan rivers) and high CDOM concentrations (at the blackwater rivers) (Martin et al., 2018); at the marine stations, this

most likely reflects the low nutrient concentrations that are typical for tropical seas.

## 3.2 FDOM compositional indices

The fluorescence index (FI) ranged from ~1.1 in the freshwater region of the Maludam River to ~1.5 for the coastal waters of the eastern region (Fig. 2(a)–(e)). In the Rajang River and eastern region, the FI showed no seasonal variation nor a clear trend with salinity: values ranged mostly between 1.4–1.55. The scatter in FI at salinity 0 in the Rajang River may reflect differences

in DOM between the distributary channels. All other rivers had lower FI than the Rajang, ranging from 1.1–1.5. In the western region, and the Maludam and Sebuyau estuaries, FI clearly increased with salinity. Seasonal variation in FI was only seen in the western region, where higher salinities in September were associated with ~0.05-unit higher FI than in March.

The humification index (HIX) showed a hockey stick-like distribution with salinity in the Eastern and western regions, with consistently high values (~0.9) until salinities of 20–25, beyond which HIX decreased rapidly to 0.6–0.7 (Fig. 2(f)–(j)). This

pattern closely follows expectations from conservative mixing, especially in the western region. Highest HIX values were found in the four blackwater rivers and the Sematan River, close to 1.0, with the Rajang River having somewhat lower values of 0.8–0.9. HIX did not decrease with salinity in the Maludam and Sebuyau estuaries, but salinities here were always below 25. Some seasonal variation was observed, with HIX in the eastern region reaching lower values in September and June than in March, and HIX in the western region also reaching lower values associated with higher salinity in September than in March.

### 3.3 Spatial distribution and characteristics of PARAFAC components

The five-component PARAFAC model explained 99.6 % of the variability between EEMs for the entire dataset. All five components (C1–C5) showed high similarity to components previously identified in various aquatic environments (Fig. 3, Table 1). Specifically, C1, C2, and C3 had emission maxima in the visible wavelength range, indicating a high contribution of conjugated fluorophores to these components (Coble, 1996; Fellman et al., 2010a). C1 exhibited an emission maximum at 440 nm with two excitation maxima at 255 nm and 330 nm, which is traditionally defined as Peak C (Coble, 1996). C2 had similar spectral characteristics to C1, but both the excitation and emission maxima exhibited slight redshifts. C3 showed a narrow excitation peak with a single UVC maximum and a broad emission peak centering at 460 nm, resembling the conventionally defined Peak A. All of C1, C2, and C3 have been widely recognized as humic/fulvic acid-like components derived from terrestrial plant litter (Stedmon et al., 2003; Stedmon and Markager, 2005a; Yamashita et al., 2015), and in the present study, they primarily represented the terrestrial humic-like DOM derived from peatlands. C4 was characterized by two excitation maxima (<250 nm and 310 nm) and a relatively narrow emission peak in the UVA region, which closely matches Peak M, traditionally defined as a marine humic-like component (Coble, 1996). This component is commonly found in marine surface waters, representing a heterotrophically reprocessed DOM fraction that is part of the autochthonous DOM pool and correlates with the presence of freshly produced, bio-labile compounds (Gonçalves-Araujo et al., 2015; Wagner et al., 2015; Yamashita et al., 2015; Osburn et al., 2016; Fellman et al., 2010a). C5 is a protein-like component, with its excitation/emission maxima in the traditionally defined Peak T (tryptophan-like) and Peak B (tyrosine-like) regions, and thus represents fresh DOM produced by phytoplankton (Coble, 1996; Stedmon and Markager, 2005b). The fluorescence maxima of all components and their potential sources investigated in previous literature are summarized in Table 1.

Components C1–C4 showed very similar distributions across our study region, with high values (0.1–4 RU) in the rivers, and strong decreases with salinity to values ≤0.01 RU at the most marine stations (Fig. 4). Blackwater rivers had consistently 5–10-fold higher values for C1–C4 than the Rajang and Sematan rivers, reflecting the far higher DOC concentrations in blackwater samples. C5, in contrast, showed consistently low values across the study region, mostly <0.2 RU, and without a clear difference between blackwater and non-blackwater rivers. C5 also did not decrease with salinity, instead remaining at relatively constant values across the entire salinity gradient. Interestingly, both C4 and C5 showed low or no correlation with chlorophyll-$a$ concentrations (Table 2), even though both components are often associated with autochthonous DOM (microbially reprocessed and fresh DOM, respectively, for C4 and C5). This lack of correlation may at least partly be explained by the limited variation in chlorophyll-$a$ concentration across our study region.

In the Samunsam, Maludam and Sematan rivers, C1, C2, and C4 showed conservative mixing with seawater (Fig. 4). C3 showed evidence of non-conservative behavior in the Maludam and at some stations in the Samunsam, but behaved conservatively in the Sematan river. In the Rajang River, C1–C4 all showed positive deviations from conservative mixing, suggesting that there were additional inputs of all of these components in the Rajang estuary. A mixing model was not

calculated for the Sebuyau River because it drains into the estuary of the larger Lupar River, for which we could not collect freshwater end-member samples.

Seasonal variation was not seen for any components in the eastern region. In the western region, seasonal differences were observed for components C1–C4: the higher salinities at the most marine stations in September were associated with lower values of all four components. Moreover, C1, C2, and C4 were higher in September (end of the southwest monsoon) in the Samunsam and Sematan rivers compared to March (end of northeast monsoon), although C3 did not differ seasonally in these two rivers. In contrast, seasonal variation was only observed for C3 in the Maludam, Sebuyau, and Simunjan rivers, all of which had consistently lower values in September than in March. C1, C2, and C4, however, showed no seasonality in these three rivers. C5 did not vary seasonally in any of the rivers.

## 3.4 Behavior of FDOM fractions during photodegradation

Martin et al. (2018) already reported the losses of DOC and CDOM observed during the photodegradation experiments, with 5.6–26 % of riverine DOC removed after 3–5 days of sunlight exposure (Fig. 5(a)–(d)). We found even greater percentage losses of the four humic-like components (C1–C4) in the two Rajang River samples and the eastern region seawater sample (Fig. 5), with C1 and C2 showing greater losses (50–6 8% reduction) than C3 and C4 (26–50 % reduction). The reduction in all four humic-like FDOM components in the seawater experiment is particularly notable (Fig. 5(g), (k), (o), (s)), because no loss of DOC was observed in this experiment (Fig. 5(c)) (Martin et al., 2018). The protein-like component, C5, showed no change relative to controls in the Rajang and seawater experiments, except for possibly a minor degree of photoproduction in the September Rajang experiment. Otherwise, no photoproduction of FDOM was observed during the Rajang and seawater experiments. Sunlight exposure caused a small decrease in HIX in the two Rajang River experiments, where a slight reduction in FI was also observed, and in the seawater experiment (when comparing light versus dark bottles in this experiment, rather than relative to the initial sample).

The Samunsam River blackwater showed reductions in C1 and C2 by the end of the experiment, but the same phenomenon was also observed for the dark control samples. One of the dark-treated samples on Day 6 was considered as an outlier and omitted due to its abnormal EEM spectra. Only C2 showed clear photodegradation in excess of the dark controls. C3 and C4 of light-exposed samples were actually elevated after one day, followed by small decreases during the subsequent days, with the data overall suggesting some degree of photoproduction of C3 and C4 in this river. C5 showed a small increase relative to the initial sample in the Samunsam experiment, but dark and light samples were within error of each other. Unlike in the other experiments, HIX and FI did not change during the Samunsam River experiment.

## 4 Discussion

### 4.1 FDOM markers as tracers of DOM sources in Sarawak

The fluorescence and humification indices (FI and HIX) are easily quantifiable markers that are commonly used to trace tDOM. In particular, FI is thought to distinguish terrestrially derived fulvic acids (FI<1.4) from microbially derived fulvic acids
(FI>1.4) (Cory et al., 2010). The FI values at mid- and low salinities of the entire study region, except the Rajang River, thus suggested that a large proportion in the total DOM pool is sourced from land. The lowest FI values were observed in the blackwater rivers (*e.g.* 1.1–1.2 in the Maludam River), consistent with the large inputs of peatland-derived DOM. Similarly low FI values (1.2–1.3) had been reported in various temperate and Arctic rivers and swamps with large terrestrial DOM input (Cory et al., 2010; Cory and McKnight, 2005; Helms et al., 2014; Mann et al., 2016). The increase in FI with salinity in the
western region, Maludam River and Sebuyau River, tracked the dilution of tDOM during estuarine mixing and the shift in the composition of the total DOM pool towards higher relative contribution from microbially derived DOM. However, we note that the ranges in FI of terrestrial *versus* microbial DOM endmembers are reported as quite variable in the literature (Cory et al., 2010; McKnight et al., 2001) and the appropriate wavelength range to use for FI calculations is also still debated: the emission wavelengths of 470 nm/520 nm proposed by Cory et al. (2010) can yield unreasonably high values (Kida et al., 2018).
Thus, the FI values of ~1.5 for the Rajang River do not necessarily indicate the dominance of microbially derived DOM over tDOM. This shows that caution is warranted when relying on simple fluorescence indices to trace tDOM.

The high HIX values in all rivers suggest a very high degree of humification of the DOM. The values in all rivers except the Rajang River were >0.9, overlapping with the range of HIX of fulvic acid extracted directly from agricultural soils (0.90–0.96) (Ohno, 2002). HIX declined in coastal waters, indicating a shift towards less humified DOM in coastal waters. Given the
wavelength ranges used to calculate HIX, we note that HIX should be very similar to the ratio of our C5 Fmax to C1 Fmax (C5/C1 Fmax ratio), thus indicating the relative proportions of allochthonous versus autochthonous DOM. Indeed, we found a significant and very strong correlation between HIX and C5/C1 Fmax ratio ($r^2 = 0.92$, $p < 0.01$, Fig. S1). This shows that HIX appear to be a robust tracer of tDOM in our study region. This conclusion is supported by the low CDOM spectral slope ($S_{275-295}$) and high SUVA$_{254}$ reported for these rivers by Martin et al. (2018). The humification process produces high-molecular
weight aromatic compounds (Zech et al., 1997), and $S_{275-295}$ and SUVA$_{254}$ are correlated with mean molecular weight (Helms et al., 2008) and with aromaticity (Weishaar et al., 2003), respectively. One might therefore expect these CDOM parameters to be closely related to HIX. Interestingly, however, HIX only showed relatively weak correlations with SUVA$_{254}$ ($r^2 = 0.58$, $p < 0.01$) and with $S_{275-295}$ ($r^2 = 0.65$, $p < 0.01$), suggesting that the HIX does not trace identical chemical properties of the organic matter as the two CDOM parameters (Fig. S1). While we can rule out significant errors in SUVA$_{254}$ and $S_{275-295}$ due
to the NaN$_3$ blanks (Supplementary Information 1), the presence of Fe(III) can lead to over-estimates in SUVA$_{254}$ (Poulin et al., 2014). Although we do not have Fe(III) measurements to quantify this potential error, nearly all of our freshwater samples have SUVA$_{254}$ of less than 5.5 with decadic absorption coefficients often exceeding 100 m$^{-1}$, suggesting that Fe(III) probably did not bias our estimates to a very great degree.

The strong similarity in spatial distribution, *i.e.* conservative mixing, between our components C1–C4 suggests that they were most likely all of terrestrial origin. This is further supported by the fact that the differences in C1–C4 values between the rivers broadly reflected their DOC concentrations, with lowest values in the Rajang and Sematan, and higher values in the blackwater rivers. Previous studies have also found that multiple terrestrial humic-like components can show similar biogeochemical behavior along the aquatic continuum within a region (Stedmon et al., 2003; Murphy et al., 2008; Yamashita et al., 2011; Gonçalves-Araujo et al., 2015). Nevertheless, our C1–C4 do very likely correspond to chemically distinct tDOM fractions. C1 and C2 share spectral characteristics that are conventionally assigned as humic compounds leached directly from soils, and typically show high photolability (McKnight et al. 2001; Stedmon et al. 2003; Lapierre and del Giorgio 2014; Yamashita et al. 2015). Our C3 has spectral characteristics that are also associated with terrestrial humic DOM, but often also indicative of moderate photochemical processing (Stedmon et al., 2007; Cawley et al., 2012). This is consistent with our experimental results that show lower photolability, and possibly even some photoproduction, of C3 compared to C1 and C2. Moderate photoproduction of C3 might explain why some samples in the western region deviated clearly from conservative mixing (Fig. 4(l)). Stubbins et al. (2014) further showed that C3 may represent highly aromatic and black carbon compounds, characterized by higher molecular weight, higher diversity in molecular structure, and depletion in nitrogen compared to C1, which matches lignin-like compounds and is less modified by reprocessing after its production from plant litter.

C4 represents another class of humic-like DOM, but C4 is conventionally assigned as a marine humic-like component, and thought to be generated by heterotrophic reprocessing of aquatic autochthonous DOM (Coble, 1996; Cory and McKnight, 2005; Fellman et al., 2010a). Higher concentrations of C4 are commonly reported in productive waters, such as coastal upwelling regions and at mid-salinities in some estuaries (Coble et al., 1998; Yamashita et al., 2008; Fellman et al., 2010b). This component can be produced by bacterial reprocessing of fresh phytoplankton-derived organic matter (Kinsey et al., 2018), but also directly by phytoplankton in the absence of bacteria (Romera-Castillo et al., 2010). However, in this study, because C4 showed such a close correlation with C1 (Spearman's $\rho$>0.898, p<0.01, Table 2), but not with chlorophyll-*a* or C5, we inferred that C4 was unlikely to be associated with aquatic primary production. Instead, C4 almost certainly had a terrestrial source from peatlands, although it is possible that our C4 is actually microbially reprocessed tDOM, as suggested by other studies (Stedmon et al., 2003; Murphy et al., 2008; Yamashita et al., 2011). In addition, our photodegradation experiment with the Samunsam water suggested that there might be some photoproduction of C4, although overall C4 showed a more conservative mixing pattern than C3 in the western region.

C5 has spectral characteristics that are generally associated with protein-like DOM, although our C5 falls in between the canonical tryptophan-like and tyrosine-like peaks (Yamashita et al., 2015). High concentrations of protein-like components are typically reported during algal blooms, and are generally thought to trace fresh, autochthonous DOM in fresh- and seawater (Stedmon and Markager, 2005; Murphy et al., 2008; Yamashita and Jaffé, 2008; Jørgensen et al., 2011). C5 is produced by phytoplankton cultures (Kinsey et al., 2018; Romera-Castillo et al., 2010), but production rates vary between phytoplankton species (Fukuzaki et al., 2014). Furthermore, Yamashita et al.(2015)found that the DOC-normalized protein-like component Fmax value is indicative of the amino acid content in DOM and thus the bioavailability of DOM. Interestingly, we found no

correlation between C5 and chlorophyll-*a* in our study region. This could be caused by several factors: for one, chlorophyll-*a* was consistently low across our study region, so there might simply not have been enough variation in aquatic primary production to cause a correlation. For another, spatial and temporal variation in phytoplankton community composition could have obscured a correlation between C5 and chlorophyll-*a* across our entire dataset. Moreover, protein-like components are typically labile to biodegradation (Wickland et al., 2007; Lønborg et al., 2010; Kinsey et al., 2018), so their production rates are not necessarily reflected in their concentrations. Finally, it has even been suggested that protein-like components can be associated with the degradation of terrigenous organic matter (Stedmon and Markager, 2005a; Yamashita et al., 2011), but the fact that our C5 did not consistently decrease with salinity ruled out a primarily terrestrial source for this component.

All our components except C2 resembled those identified recently in the Kinabatangan River in northeast Borneo, the catchment of which consists of oil palm plantations and natural forests (Harun et al., 2016), suggesting a relatively similar organic matter composition across coastal Borneo. Harun et al. (2016) showed clear seasonal variations, with higher concentration of peak A, which dominated their FDOM pool, in the wet season relative to the dry and inter-monsoonal season. This is similar to the seasonal difference in C3 in our blackwater rivers. Harun et al. (2016) also inferred an anthropogenic source of peak M from land use change and highlighted the importance of microbial and/or photochemical processing of tDOM to its production, supporting our interpretation of a terrestrial source for C4 with heterotrophic reworking.

## 4.2 Photochemical transformations of FDOM

We observed high photolability of the four terrestrial components (C1–C4) in the Rajang River and seawater samples, with percentage losses of the FDOM components that substantially exceeded the loss in DOC. Moreover, as suggested by Helms et al. (2014), the decrease in HIX indicated a change to an overall less humified DOM pool with preferential losses of aromatic compounds in these three experiments. We note that although sunlight irradiation can cause spectral shifts instead of complete loss of fluorescence (Helms et al., 2013), examination of our excitation and emission spectra showed large decreases in fluorescence intensity, but no shift of spectral peaks (Fig. S2). Large losses of terrestrial humic components, changes in CDOM spectra, and reductions in molecular markers such as lignin phenols are commonly reported from photodegradation experiments with aquatic samples (Stedmon et al., 2007; Spencer et al., 2009; Stubbins et al., 2010). However, studies in some environments have also reported very limited tDOM photolability (Chupakova et al., 2018; Stubbins et al., 2017), highlighting the need for more experiments.

Interestingly, the Samunsam River water showed less pronounced photodegradation of FDOM components, despite experiencing the greatest photomineralization of DOC. The fact that HIX did not change in this experiment can be explained by the photoproduction of C3, which would have offset the decline in C1 and C2. It is unclear why the FDOM components in the Samunsam water showed more limited photodegradation, given the large loss of CDOM and changes in CDOM spectral slopes (Martin et al., 2018), but these data may suggest a degree of variation between rivers in photolability and possibly in chemical composition of our FDOM components.

The protein-like component (C5) was photoresistant in all experiments, indicating low photolability of autochthonous DOM. Differences in photolability between DOM fractions are usually linked to the relative proportions of aromatic (more photolabile) versus aliphatic (less photolabile) structures (Helms et al., 2014; Stubbins et al., 2010), and phytoplankton-derived organic matter is generally dominated by more aliphatic compounds such as carbohydrates, proteins, and lipids ( Lancelot, 1984).

## 4.3 FDOM-based estimate of terrigenous DOC fraction

Estimates of the proportion of tDOC in marine environments have been based mostly on C/N ratio, isotopic composition, and biomarkers such as lignin; such studies have shown that tDOC accounts for 0.5 %–2.4 % of total DOC in the open Pacific and Atlantic Oceans (Meyers-Schulte and Hedges, 1986; Opsahl and Benner, 1997), 5 %–22 % in the Arctic shelf seas (Opsahl et al., 1999), and ≤30 % on the Louisiana Shelf (Fichot and Benner, 2012). These analyses are relatively laborious and expensive. However, given that fluorescence analysis can distinguish between terrigenous and autochthonous fractions, FDOM might hold the potential to estimate tDOC in certain environments, provided that both FDOM and the bulk DOC pool mix conservatively with at most minor biogeochemical modifications. Terrestrial humic-like PARAFAC components have been shown to be strongly correlated with lignin phenol concentrations in various aquatic environments (Stedmon et al., 2003; Walker et al., 2009; Yamashita et al., 2015). In particular, C1 has been widely recognized as a component representing high molecular weight, humic-like degradation products of lignin (Coble 1996; McKnight et al., 2001; Stedmon et al., 2003; Stubbins et al., 2014). C1 correlates particularly strongly with lignin phenols (Yamashita et al., 2015), and is detected only in trace amount in the open oceans, *e.g.* ~0.006 R.U. in the tropical Atlantic Ocean (Murphy et al., 2008). This suggests that C1 can potentially be used as a tDOM tracer, if we assume that C1 behaves biogeochemically in approximately the same way as the total (fluorescent and non-fluorescent) tDOM pool while the marine endmember contributes no C1. We therefore attempted to estimate tDOC in our coastal samples from the ratio of C1 Fmax to DOC (Eq. 3):

$$\%tDOC = 100\ \% \times \frac{(C1\ Fmax/DOC)sample}{(C1\ Fmax/DOC)river}\ (3)$$

where %tDOC is the percentage contribution of tDOC to the total DOC pool, and (C1 Fmax/DOC)sample is the DOC-normalized C1 in the sample for which %tDOC is to be estimated, and (C1 Fmax/DOC)river is the highest value of DOC-normalized C1 Fmax at salinity 0 within the appropriate river, as the riverine endmember. This is the most conservative way of selecting the riverine endmember value to avoid overestimating %tDOC in marine samples, but if we use the mean freshwater C1/DOC value for each river, our final %tDOC estimates are only ≤4 percentage points higher. For the eastern region samples, we used the Rajang River as the riverine endmember. For the western region samples, the Samunsam River served as the riverine endmember due to its likely larger DOC export compared to the Sematan. To calculate the uncertainty for our estimate, we propagated the uncertainty in DOC (±4.3 %, Martin et al. (2018)) and in Fmax values (±1 %, Korak et al. (2014)), which yielded an uncertainty of around ±6 % in the final %tDOC estimate. Our analytical uncertainties are thus very minor.

The %tDOC generally decreased with salinity and reached minimum values of 15±0.9 % –25±1.5 % at stations with highest salinity in both regions (Fig. 6), consistent with the low HIX at these stations. Interestingly, %tDOC exceeded 100 % at a few

mid-salinity stations in the western region. This could indicate that the freshwater endmember C1/DOC ratio for the Samunsam River was underestimated; only a single freshwater sample could be collected in each season from this river. Alternatively, there could be additional sources of C1-rich DOM within the Samunsam estuary, perhaps either from the surrounding mangrove vegetation or as a result of resuspension of sediments that might sorb and de-sorb tDOM. The Samunsam estuary was sampled as strong tidal currents were visibly causing strong resuspension. This issue calls for further work to investigate the use of FDOM as a quantitative tracer of tDOC.

Our FDOM-based estimate needs to be viewed with caution, since we cannot fully test the underlying assumptions. Our approach assumes firstly that C1 is exclusively terrestrially derived, and has no non-terrestrial sources in estuaries and marine waters. Secondly, the approach assumes that C1 behaves biogeochemically in approximately the same way as the total (fluorescent and non-fluorescent) tDOM pool (Wagner et al., 2015). The first assumption is probably broadly valid: as discussed above, Fmax values of C1-like components in open-ocean waters are very low relative to the values across our study area. The second assumption is probably not seriously violated in our study, since we observed close to conservative mixing of both DOC and C1 in our region. In fact, the most likely degradation process we have identified, i.e. photodegradation, potentially causes preferential loss of C1 relative to total DOC, which would actually cause us to underestimate the true %tDOC in marine samples. However, over the relatively short spatial scales over which we sampled, the residence time of tDOM is probably short relative to the rates of biogeochemical tDOM processing, such that our estimates of %tDOC are perhaps not impacted strongly by any differences in degradation rates of C1 *versus* bulk tDOC. The relatively high tDOC contribution that we estimate for our coastal stations is also in the same range as estimates in other tDOM-influenced regions (Fichot and Benner, 2012; Opsahl et al., 1999), suggesting that our estimates are plausible. Moreover, we found a close exponential relationship between %tDOC and $S_{275-295}$ (%tDOC = exp $(\alpha + \beta S_{275-295})$, where $\alpha=1.48$, $\beta=-126.23$, Fig. S1(b)), similar to the exponential relationship between %tDOC as estimated from lignin phenols and $S_{275-295}$ shown by Fichot and Benner (2012). A relatively high tDOC contribution to the coastal DOC pool is also consistent with our finding that marine waters still contained photolabile terrestrial FDOM components, and showed increases in CDOM spectral slope compared to river waters (Martin et al., 2018).

## 4.4 Biogeochemical fate of tDOM in Sarawak

All of our terrestrial FDOM components, C1–C4, displayed mostly conservative mixing with seawater, which suggests that tDOM does not undergo major biogeochemical processing in the rivers and estuaries. The same conclusion was also reached by Martin et al. (2018) based on the distribution of DOC and CDOM parameters. The fact that our fluorescence data independently show very similar results increases our confidence in this conclusion. The main exception to this pattern was observed in the Rajang River delta, where C1–C4 consistently showed higher values in the estuary than expected from conservative mixing. Based on the DOC distribution in the delta, Martin et al. (2018) hypothesized that this reflected DOC input from surrounding peatlands, even though the concomitant increase in $S_{275-295}$ did not unambiguously support a terrigenous

origin of this DOC. The fact that we see the same positive deviation from conservative mixing in all four terrestrial components, but not in our C5, strongly supports the idea that the additional DOC input into the Rajang River distributaries consists of tDOC from the peatlands, and not from autochthonous production.

We inferred in this study that C4 was terrestrial, as also shown by Harun et al. (2016) in northeastern Borneo. This suggests that in Southeast Asia, Peak M might not be part of the autochthonous marine DOM pool. Because microbial processing plays a major role in soil organic matter transformation within peatlands, we hypothesize that C4 is produced within the soil prior to the export of tDOM to rivers. The conservative mixing behavior of C4 rules out significant production by heterotrophic processing of tDOM within rivers and estuaries.

Our experimental results shed further light on the biogeochemical fate of tDOM in this region by showing the high degree of photolability of terrestrial FDOM in Sarawak. The predominantly conservative mixing of our terrestrial FDOM components thus further indicates that substantial biogeochemical processing of tDOM probably only takes place once it has mixed into marine waters with greater light penetration. This contrasts, for example, with results from the Mississippi estuary, where preferential removal of high-molecular weight compounds and oxidation of lignin were reported at the boundary from mid- to high-salinity waters, mostly as a result of photooxidation (Hernes and Benner, 2003).

**Conclusions**

Tropical peatlands in Sarawak, Borneo, export extremely humified DOM to coastal waters. We have identified four terrestrial humic-like PARAFAC components (C1–C4) that have high concentrations in peat-draining rivers, and mix conservatively with seawater. The rivers are dominated by terrigenous DOM, and we estimated that even our marine stations are characterized by relatively high tDOM concentrations. One FDOM compositional index, the HIX, yielded results consistent with our PARAFAC analysis and thus can serve as a robust tracer of tDOM in coastal Sarawak, but the FI yielded more ambiguous results. Moreover, we found no evidence of genuinely marine-produced humic substances, with the canonical marine humic component also tracing terrestrial input. Although our experimental evidence shows high photolability of terrestrial FDOM, our observational data suggest that tDOM in Sarawak experiences little biogeochemical processing until it reaches fully marine waters.

**Code and data availability**

The full processed data set is available as the Supplementary Data Table 1. All raw data files, MATLAB scripts for EEM correction and PARAFAC analysis are available through the NTU data repository, DR-NTU, at https://doi:10.21979/N9/RCYCIT (Zhou, 2019) and https://doi.org/10.21979/N9/0RLSDU (Martin, 2018).

## Author contribution

YZ processed and analyzed the data. PM and MM conceptualized the research project, obtained research funding, and led the field sampling and sample measurements. All authors contributed to the writing of the manuscript.

## Competing interests

The authors declare that they have no conflict of interest.

## Special issue statement

This article is part of the special issue "Biogeochemical processes in highly dynamic peat-draining rivers and estuaries in Borneo". It is not associated with a conference.

## Acknowledgements

Research permits were granted by the Sarawak Forestry Department and the Sarawak Biodiversity Centre (permit number: NPW.907.4.4(Jld.14)-161, Park Permit No WL83/2017, and SBC-RA-0097-MM). We are indebted to the boatmen who helped us collect samples: Lukas Chin, Captain Juble, and their crew (Rajang river and eastern region), and Minhad and Pak Mat (western region). We thank Claire Evans, Joost Brandsma, and Aazani Mujahid for help in planning and leading part of the field work, and Ashleen Tan Su Ying for collecting and measuring most of the FDOM samples. Faddrine Jang, Edwin Sia,
Gonzalo Carrasco, Jack Sim, Akhmetzada Kargazhanov, Florina Richard, Faith Chaya, Noor Iskandar Noor Azhar, and Fakharuddin Muhamad provided essential logistical support in the field. We thank Colin Stedmon, Kathleen Murphy, and Urban Wünsch for assistance with PARAFAC analysis during the 2018 Organic Matter Fluorescence Spectroscopy Workshop in Copenhagen, Denmark. Amanda Cheong Yee Lin helped with the PARAFAC analysis. We thank the two reviewers and the associate editor for constructive criticism that improved the original manuscript. P.M. was funded through a Tier 1 grant from
the Singapore Ministry of Education's Academic Research Fund (RG 175/16).

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

30

## Tables and Figures

**Table 1.** The excitation and emission maxima of our PARAFAC components, and their possible sources and corresponding chemical compounds (wavelengths in brackets are secondary maxima). The Tucker congruence coefficients (TCC) are always above 0.95, indicating strong correlations. The respective TCC values can be found in the Supplementary Data Table 2.

| Component | Excitation maxima | Emission maxima | Possible source/ classes of compound |
|---|---|---|---|
| C1 | 330 (255) | 440 | terrestrially derived humic matter[1,2] with high molecular weight[3] degraded from lignin[4,11] |
| C2 | 275 (385) | 506 | soil fulvic acid[5,6,7], reduced semi-quinone fluorophore derived from terrestrial higher plants and associated with microbial reduction reactions[8] |
| C3 | <250 | 460 | Terrestrially derived humic matter[1,2,3,5], photo-product[12], aromatic and black carbon compounds with high molecular weight and depleted of N[11] |
| C4 | <250 (310) | 390 | marine humic-like, microbially processed autochthonous compound[1,7,8,9] |
| C5 | 275 | 328 | protein, mixture of tryptophan-type and tyrosine-type compounds, autochthonous DOM[1,2,10] |

([1]Coble, 1996; [2]Yamashita et al., 2015; [3]Stedmon et al., 2003; [4]McKnight et al., 2001; [5]Stedmon and Markager, 2005a; [6]Lochmüller and Saavedra, 1986; [7]Yamashita and Jaffé 2008; [8]Cory and McKnight, 2005; [9]Fellman et al., 2010a; [10]Yamashita et al., 2010; [11]Stubbins et al., 2014; [12]Stedmon et al., 2007)

**Table 2.** Spearman's rank correlation between PARAFAC components (C1, C4 and C5) and chlorophyll-a concentrations in different regions.

| | region | chlorophyll-$a$ (mg/L) | | C1 Fmax (R.U.) | |
|---|---|---|---|---|---|
| | | $\rho$ | Sig. | $\rho$ | Sig. |
| C4 Fmax | Eastern | **−0.370** | **<0.01** | **0.988** | **<0.01** |
| (R.U.) | Western | 0.019 | 0.897 | **0.972** | **<0.01** |
| | Blackwater | -0.304 | 0.053 | **0.898** | **<0.01** |
| C5 Fmax | Eastern | **−0.363** | **<0.01** | | |
| (R.U.) | Western | -0.028 | 0.850 | | |
| | Blackwater | -0.178 | 0.266 | | |

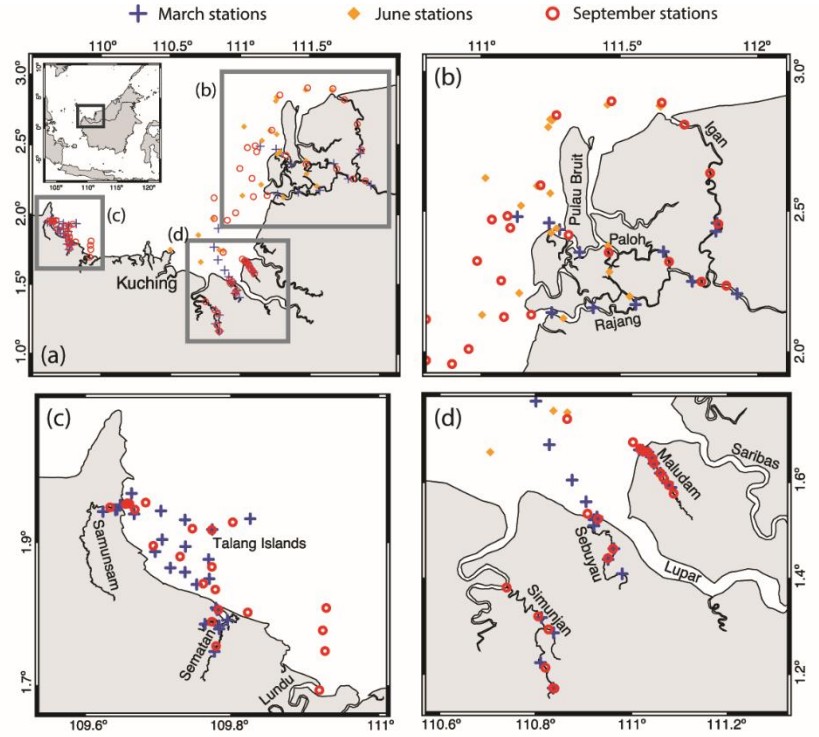

**Figure 1.** (a) Map of the study region and sampling sites (Martin et al., 2018). Zooming in of the three regions is shown in panels (b) – (d).

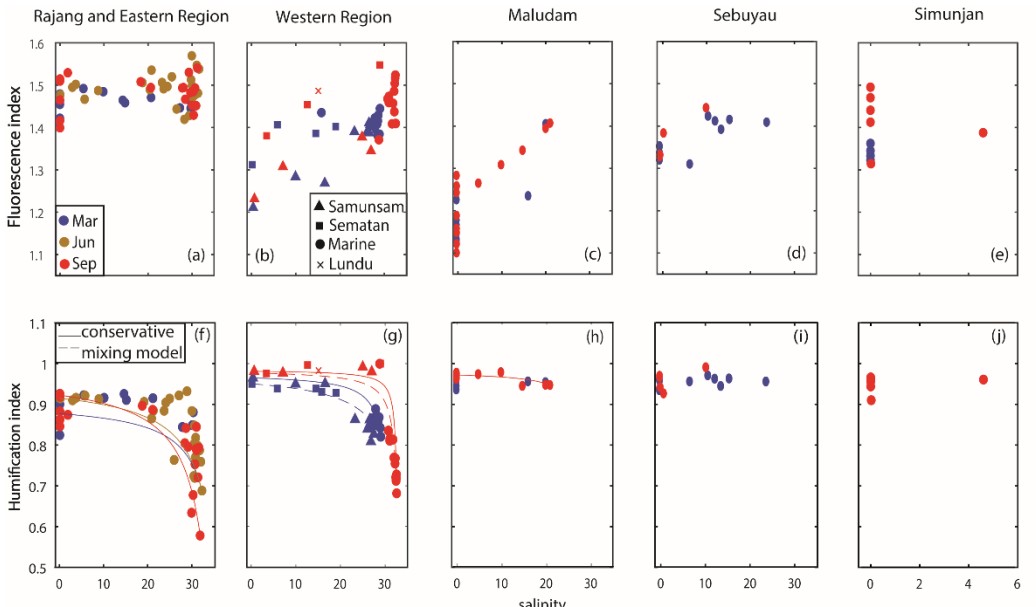

**Figure 2.** Spatial distribution of fluorescence index (FI) (a–e) and humification index (HIX) (h–l). Samples from different seasons are distinguished by colors. Samples from different regions are shown in individual panels, specified by the titles of each panel. The conservative mixing models of HIX are delineated for the Rajang and eastern region by solid lines in panel (f), for Samunsam river by solid lines and for Sematan river by dashed lines in panel (g), and for Maludam river by the solid line in panel (h).

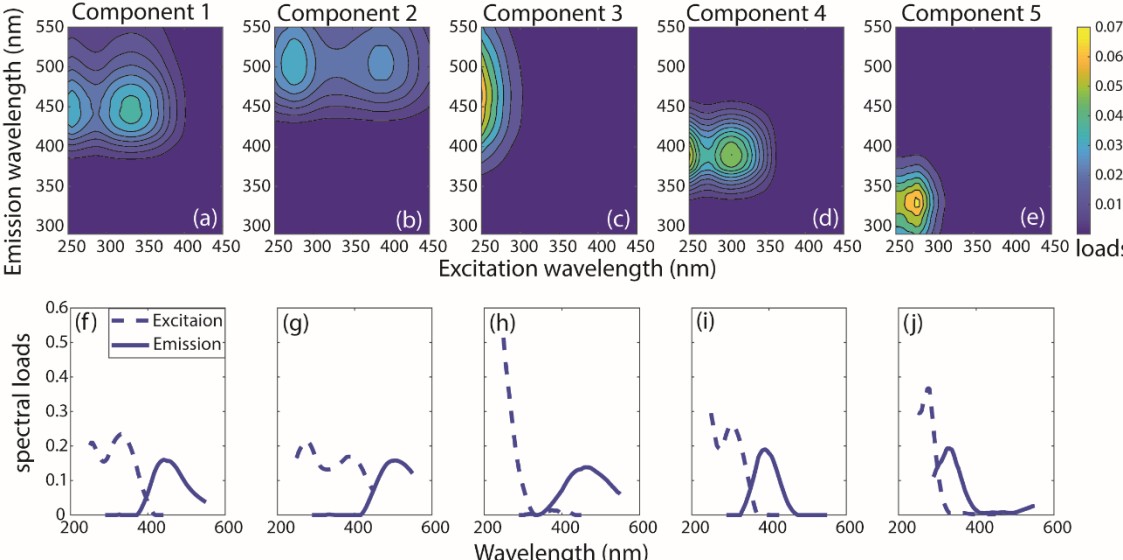

**Figure 3.** The 3-D fingerprint spectra (a–e) and spectral loads (f–j) of the five components identified by PARAFAC analysis. The overlaid excitation and emission loadings of the validated split dataset can be found in Figure S4.

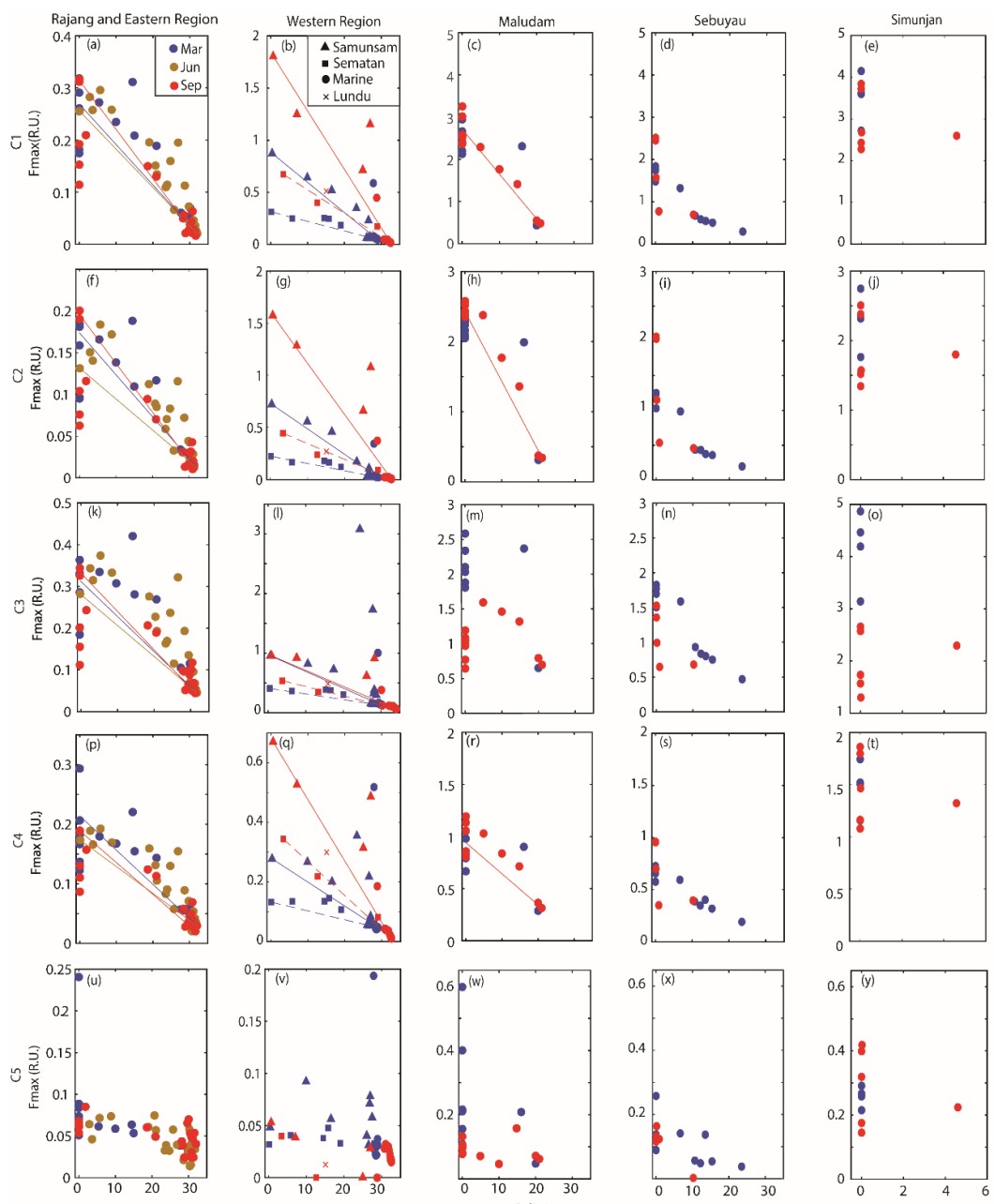

**Figure 4.** The spatial distribution of  C1–C5 Fmax (a–y) for the Rajang and eastern region, the western region, Maludam River, Sebuyau River and Simunjan River. Colors distinguish samples from different seasons in panel (a) to (y). The conservative mixing models of C1–C4 are delineated for the Rajang and eastern region by solid lines, for Samunsam River by solid lines, for Sematan River by dashed lines, and for Maludam River by solid lines.

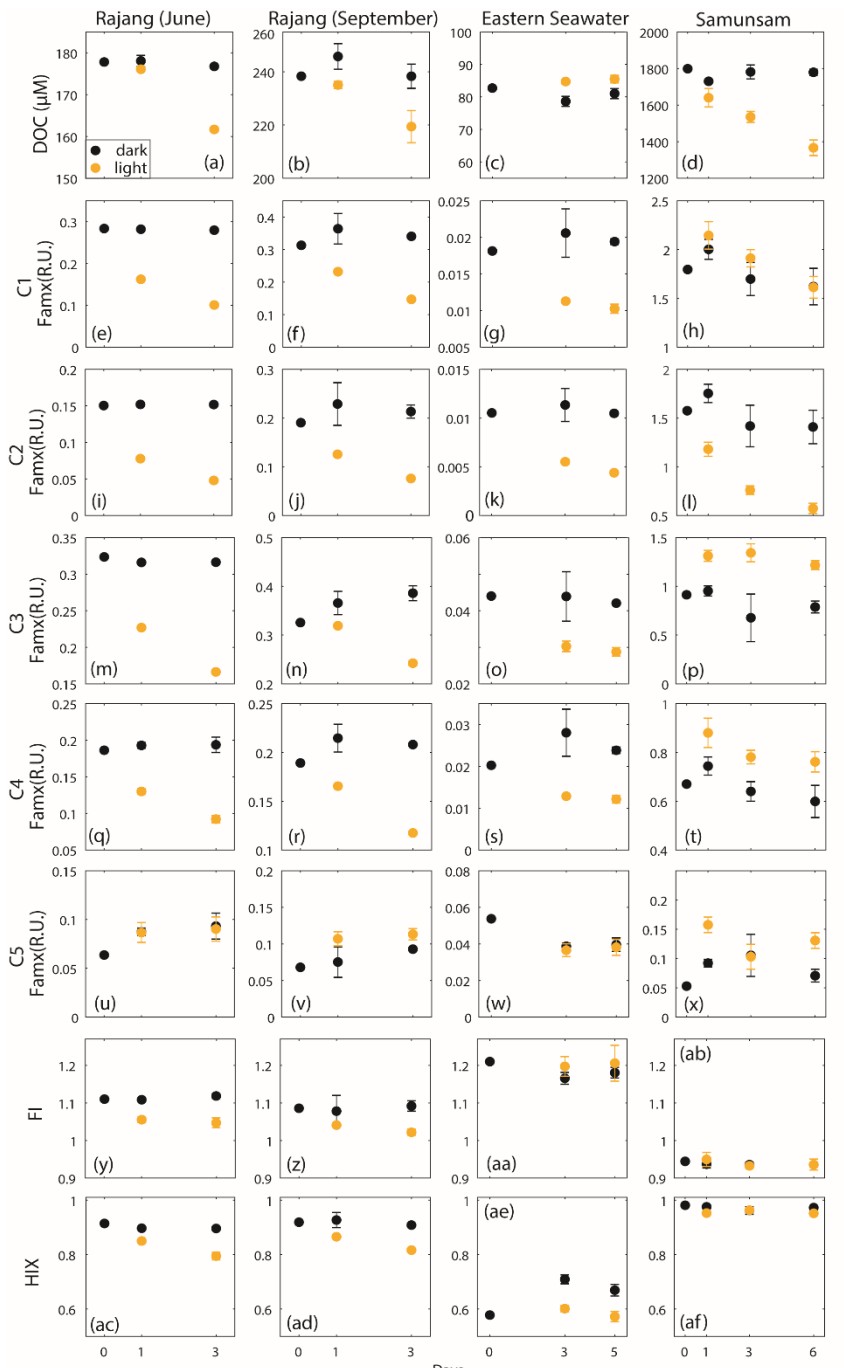

**Figure 5.** Changes in DOC (a–d), C1–C5 Fmax (e–x), FI (y–ab) and HIX (ac–af) of samples from Rajang River in June and September, from seawater of eastern region and from Samunsam River during the photodegradation experiment. DOC data are taken from Martin et al. (2018).

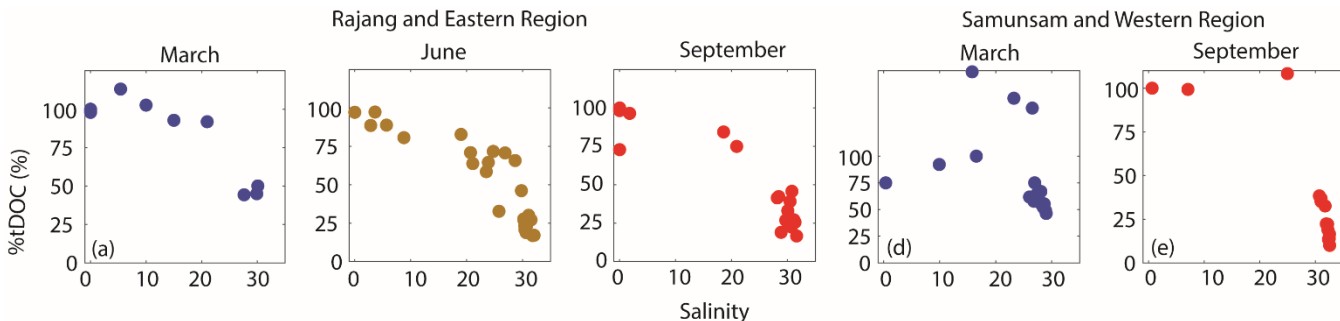

**Figure 6.** Estimated percentage contribution of terrigenous DOC to the total DOC pool (%tDOC) against salinity for all estuarine samples in the eastern and western regions.