# Peer review of "Composition and cycling of dissolved organic matter from tropical peatlands of coastal Sarawak, Borneo, revealed by fluorescence spectroscopy and PARAFAC analysis"

_Biogeosciences, 2018_

## Referee Comment (RC1) · Kida (Referee) · 3 Jan 2019

**Original Submission**

Ms. Ref. No.: bg-2018-508

Title: Composition and cycling of dissolved organic matter from tropical peatlands of coastal Sarawak, Borneo, revealed by fluorescence spectroscopy and PARAFAC analysis

Authors: Yongli Zhou, Patrick Martin, Moritz Müller

**Overview and general recommendation**

**1. Recommendation**

Major revision

**2. Comments to Author**

Note: I also read the companion paper 'Distribution and cycling of terrigenous dissolved organic carbon in peatland-draining rivers and coastal waters of Sarawak, Borneo' and the relevant review comments and the authors' answers to the comments.

This study (bg-2018-508) aimed to distinguish different fractions of dissolved organic matter (DOM) in peat-draining rivers, estuaries, and coastal waters of Sarawak, Borneo, using fluorescence spectroscopy and parallel factor (PARAFAC) analysis. The authors observed that the terrigenous fractions showed high concentrations at freshwater stations within the rivers, and conservative mixing with seawater across the estuaries, while the autochthonous DOM fraction showed low concentrations at all salinities. The authors claim that, based on the fluorescence data and little changes in optical properties of DOM, at least 20%–25% of the DOC at even the most marine stations (salinity >31) was terrestrial in origin. Although not all of the data provided is new to the relevant field, the content of this paper fulfills the requirements for the submission to *Biogeosciences* of which aims and scopes are to publish studies *on all aspects of the interactions between the biological, chemical, and physical processes in terrestrial or extraterrestrial life with the geosphere, hydrosphere, and atmosphere*. The title is representative of the article contents and the abstract summarize the contents clearly. Therefore, I recommend accepting this paper after the authors revise all the necessary points.

I have serious concerns about the use of sodium azide ($NaN_3$) as a preservative for samples analyzed by UV absorption and fluorescence spectroscopy. Also, the emission wavelengths used to calculate fluorescence index (FI) seem inadequate. Finally, estimation of %tDOM by fluorescence is questionable.

**2.1 Major comments**

2.1.1. Estimate of terrestrial contribution

FDOM is only a small portion of the bulk DOM, and thus estimation of %tDOM by fluorescence is troublesome. PARAFAC components can be used to better understand biogeochemical processes that occur during the estuarine mixing, but PARAFAC components alone are not sufficient to estimate the tDOM contribution at given salinity. To make it possible, you must assume that all the rest of components in riverine DOM other than FDOM (PARAFAC component C1 in this case) behaves in the same way as C1 does during the estuarine mixing and that marine end-member has no C1. Please explicitly state your assumptions. It's not enough in the current form. Also, how do you explain %tDOC of >100% in Samunsam and Western Region (in March) at salinity >10 under your assumption?

In addition, Fmax/DOC is known to be susceptible to errors caused by the fluorescence intensity and DOC measurements (Korak et al. 2014), and the authors should include an evaluation of such an uncertainty (error propagation analysis), since %tDOM estimation is I believe the most important part of this study.

Generally, in estuarine environments, contribution from estuarine vegetation (mangrove and marsh) is done by an end-member mixing model (0.1 salinity increment) using DOC concentrations of the fresh and marine end-member (Cawley et al. 2014). Because the main subject of this study is tropical peatlands, I feel that what the authors want to investigate is not riverine (derived from upper regions) inputs but inputs from the peatlands located in the estuary. The authors may reassess contributions from the peatlands using the method reported in, for example, Cawley et al. (2014).

2.1.2. NaN$_3$

Although you said 'NaN$_3$ did not contribute any blank fluorescence', it did contribute to sample absorbance, as you mentioned in the companion paper. Indeed sample preservation is still a major challenge, and I do use NaN$_3$ to preserve samples for DOC analysis. However, I never use NaN$_3$ to preserve samples for optical analysis because of the strong UV absorbance by NaN$_3$ even at a low concentration (0.005% (w/v) in this study). I agree that if your samples have high absorbance, you could correct for the NaN$_3$ absorbance accurately. However, when measuring EEM for samples containing NaN$_3$, it seems that you failed to correct for the inner-filter effects (IFEs) caused by NaN$_3$, because for the IFEs correction you used the absorbance of CDOM that were obtained by subtracting the absorbance of NaN$_3$ from that of samples containing NaN$_3$. In that way, you underestimated fluorescence in the EEM regions where NaN$_3$ absorbed light (Ex 250–280). This is very serious because you mentioned the protein-like component 'showed consistently low values across the study region', and this could be due to underestimation of the protein-like component. The relative degree of the underestimation will be larger with decreasing sample absorbance relative to that of NaN$_3$.

If you will correct (or may have corrected) for IFEs including NaN$_3$ absorbance, please

explain the degree of uncertainty of the correction. Because, although you said all samples had the same $NaN_3$ concentration, there should be some variation in the concentration caused by, for example, repetitive volumetric measurements of samples (30 mL) and $NaN_3$ solution (150 µL).

**2.1.3. FI**

Did you apply instrument-specific correction for EEM? If so, the emission wavelength for FI must be 470/520 nm instead of 450/500 nm (Cory et al. 2010; Kida et al. 2018), because the emission peak often lies between 450 and 500 nm when the correction applied, which makes FI meaningless (FI must be calculated on the right side of the emission peak). If not, please write so in M&M section, because in that case your results are not directly comparable with other studies. It is often observed that if not corrected for the instrument-specific bias, the variability of FI between instruments is large for a given sample.

**2.2 Minor comments**

Table 2. Was the distribution of the PARAFAC components and chlorophyll-a normally distributed? If not, Spearman's rank correlation should be used instead. Note that strong parametric linear relationships between PARAFAC components are unlikely considering the theory of PARAFAC. If components have a strong linear correlation, PARAFAC cannot resolve these components and they appear as a single combined component. Correlations between PARAFAC components are generally expressed by a log-log plot or Fmax/DOC plot (Stedmon and Markager 2005).

P2L26 'extremely high DOC concentrations' Please specify the DOC range, as it depends on person when a DOC concentration is 'extremely' high.

P3L11 Sampling
How was the weather on the sampling days? In addition to seasonal changes, daily changes in rainfall and water flow conditions would affect DOM concentrations and compositions. If you discuss seasonal changes, at least the weather should be the same.

P3L30 Was the condition of the photodegradation experiment sterile (biodegradation-free)? If not, how about the effect of biodegradation? Please add some more details about the photodegradation experiment. For example, water inside the bottles was repetitively sub-sampled or you prepared many bottles and each bottle was collected as a sub-sample?

P4L3 'To minimize self-quenching of fluorescence intensity' Please add information on the maximum absorbance value of the measured samples, since IFE correction becomes invalid if sample absorbance

is too high. Also, how you measured absorbance data is completely lacking. Please explain it in this section, and reference to the companion paper alone is not sufficient.

P4L27 'chemical compound classes' The authors need to be careful here. What PARAFAC can do is to statistically deconvolute EEMs into underlying building blocks, termed 'components', and these components are rarely related to specific chemical compounds. I think the authors understood that, but for those who are not familiar with PARAFAC, the author's statement may be misleading.

P4L28 Specify how many samples were removed.

P4L29 Please add in Fig. 3 the excitation and emission loadings of the validated split dataset.

P5L1 Fmax is not just a score value. "Fmax is calculated by multiplying the maximum excitation loading and maximum emission loading for each component by its score, producing intensities in the same measurement scale as the original EEMs" (Murphy et al. 2013).

      Also, Fmax cannot be a major of the concentration of each component *in a sample*, "because different fluorophores can have very different efficiencies at absorbing and converting incident radiation to fluorescence (Murphy et al. 2013)." Rather, "Quantitative and qualitative information may however be obtained from changes in the intensity of a given component, or in the ratios of any two components, *between samples in the dataset* (Murphy et al. 2013)."

P5L8&L15 $a_{350}$, $S_{275-295}$, $S_R$, and $SUVA_{254}$ appeared for the first time here without explanations what they are. This is not kind for those who are not familiar with the optical indices. This is relevant to my comment on P4L3. Now I think that you need to make another section in M&M that explains the absorbance measurement and absorbance-based indices. However, personally I think that you can completely cut the sentences with respect to $S_R$, $a_{350}$ and $SUVA_{254}$ since you mentioned about $S_R$ and $a_{350}$ only once or twice and did not discuss SUVA results (just correlation with HIX). As for $S_{275-295}$, you may want to use it to support your idea that an FDOM-based estimate of tDOM is OK. However, I am not totally convinced that being correlated with $S_{275-295}$ supports the correctness of your fluorescence-based %tDOM (P11L24), because estimations of %tDOM based on $S_{275-295}$ is non-linear (Fichot and Benner 2012).

P5L16 "$SUVA_{254}$, 3.08–6.89" SUVA value of 6.89 is too high. Even the highest aromaticity sample (Ar >40%) in Weishaar et al. (2003) had the SUVA value of 5.3, and the possible maximum SUVA value (~5) has been recently suggested from a molecular analysis (Kellerman et al. 2018). Iron(III) is most probably interfering with SUVA determination in your sample dataset (Poulin et al. 2014; Kida

et al. 2018). If the authors did not measure Fe(III) and also have no stored sample for Fe(III) measurement, please state in the manuscript that some of SUVA values in this study was overestimated by interferences from Fe(III) to an unknown degree. Note that, if Fe(III) contributes to $SUVA_{254}$ to a similar degree for all the samples, $SUVA_{254}$ and $SUVA_{280}$ would still have a high correlation.

Another possibility is the interference by $NaN_3$ even after the blank correction. This is possible when the sample CDOM absorbance was low. Please add the information on the $NaN_3$ absorbance contribution to sample absorbance at 254 nm. According to Fig S1&2 of the companion paper, decadic absorption coefficient of the $NaN_3$ solution was about 4 $m^{-1}$, which was about 10%–30% of that of Rajang, Sematan, and Lundu samples and 50%–200% of marine samples. These values are not trivial.

P6L3 Please add seasonal climatic information (dry? rainy?) after months so that readers can easily understand climatic conditions, not only in the M&M section.

P9L31 "correlating strongly with DOC-normalized amino acid yields" This is not a correct citation. The correlation coefficient was r = 0.62 (Fig. 8b in Yamashita et al., 2015), at best moderate correlation.

**2.3 Technical corrections**

P2L5 & L7 '0.2-0.25 Pg C $yr^{-1}$' and '40% - 50%' should be 0.2–0.25 Pg C $yr^{-1}$ and 40–50% (or 40%–50%). Please check the usage for minus ($-$), hyphen (-), en dash (–), and em dash (—). I did not correct for the rest of the manuscript.

In Fig 2&4, it would be better to set the x axis to the same scale (maximum salinity of 35) except for the Simunjan River results so that comparisons between rivers become easier and more straightforward.

The caption of Fig. 4 says 'while they distinguish samples from different regions in the panel (z)', but I can't find the panel (z).

In Table 1, pleased add Tucker congruence coefficient (TCC) values so that readers can evaluate how much the comparisons are quantitative. Add the relevant explanations in M&M section as well.

In reference list, please add a space between references to improve visibility. It's OK not to do it this time but I'm suggesting this for future reviewers.

**References**

Cawley KM, Yamashita Y, Maie N, Jaffé R (2014) Using optical properties to quantify fringe mangrove

inputs to the dissolved organic matter (DOM) pool in a subtropical estuary. Estuaries and Coasts 37:399–410. doi: 10.1007/s12237-013-9681-5

Cory RM, Miller MP, McKnight DM, et al (2010) Effect of instrument-specific response on the analysis of fulvic acid fluorescence spectra. Limnol Oceanogr Methods 8:67–78. doi: 10.4319/lom.2010.8.67

Fichot G, Benner R (2012) The spectral slope coefficient of chromophoric dissolved organic matter ( S 275 – 295 ) as a tracer of terrigenous dissolved organic carbon in river-influenced ocean margins. 57:1453–1466. doi: 10.4319/lo.2012.57.5.1453

Kellerman AM, Podgorski DC, Aiken GR, et al (2018) Unifying Concepts Linking Dissolved Organic Matter Composition to Persistence in Aquatic Ecosystems. doi: 10.1021/acs.est.7b05513

Kida M, Fujitake N, Suchewaboripont V, et al (2018) Contribution of humic substances to dissolved organic matter optical properties and iron mobilization. Aquat Sci 80:26. doi: 10.1007/s00027-018-0578-z

Korak J a, Dotson AD, Summers RS, Rosario-Ortiz FL (2014) Critical analysis of commonly used fluorescence metrics to characterize dissolved organic matter. Water Res 49:327–38. doi: 10.1016/j.watres.2013.11.025

Murphy KR, Stedmon CA, Graeber D, Bro R (2013) Fluorescence spectroscopy and multi-way techniques. PARAFAC. Anal Methods 5:6557. doi: 10.1039/c3ay41160e

Poulin BA, Ryan JN, Aiken GR (2014) Effects of Iron on Optical Properties of Dissolved Organic Matter. Environ Sci Technol 48:10098–10106. doi: 10.1021/es502670r

Stedmon CA, Markager S (2005) Resolving the variability in dissolved organic matter fluorescence in a temperate estuary and its catchment using PARAFAC analysis. Limnol Oceanogr 50:686–697. doi: 10.4319/lo.2005.50.2.0686

---

## Referee Comment (RC2) · Anonymous Referee #2 · 11 Jan 2019

In this manuscript, Zhou et al. use fluorescence spectroscopy to track the fate of peat-derived DOC through an estuary. They use a mixing model based on the fluorescence spectra to quantitatively estimate peat-derived DOM in the estuary, and show that a major fraction is peat-derived in all samples. Overall, the manuscript is well-written and most conclusions are well-supported. However, I have 2 major concerns that should be addressed before publication:

1. A major portion of the findings (high photolability of tDOM; large tDOM contribution to the shelf DOM pool) echo findings in the companion paper (Martin et al. 2018).

[Figure]

I suggest reframing the introduction and adding a paragraph briefly summarizing the findings of the companion manuscript and describing how the present study will build on this work. In particular, what you can learn from EEMs that hasn't been revealed with bulk DOC and CDOM analysis.

2. The calculation of tDOM appears to be oversimplified.

-Why is the sample with the highest value of normalized C1 Fmax used for the river endmember? Since there appears to be a lot of variation at 0 salinity, wouldn't an average be more appropriate? Is it possible to do a formal sensitivity analysis based on different choices of endmember?

-Equation 3 does not include a marine endmember, which implies that (1) C1 Fmax varies linearly with tDOC, and (2) C1 would be 0 in a hypothetical pure marine DOM sample. Both assumptions should be stated and justified. It is also assumed that C1 has the same reactivity as bulk tDOM despite representing a small, compositionally distinct fraction.

-The identification of endmembers in Table S1 doesn't match the description in the text. There are marine end-members identified (not used in Eq 3), and some of the river endmembers are presented as an average of multiple stations instead of the station with the highest C1 Fmax as indicated in the text.

-Calculation of the %tDOC should be included in the methods section and more information provided.

Other minor comments:

Methods: Photodegradation experiments should be in separate subsection

Page 9, lines 13-24: paragraph mostly repeats info found elsewhere in the paper

Line 22: moreover is not the correct word here

Page 10, lines 5-6: this sentence is contradictory

Fig 4: legend indicates colors indicate regions in panel z.. figure appears to only go to panel y

[Figure]

---

## Author Comment (AC1) · 11 Feb 2019

The responses to the Reviewer Comment 1 are posted in the supplemtary pdf file compressed in the zip file.

Please also note the supplement to this comment:
https://www.biogeosciences-discuss.net/bg-2018-508/bg-2018-508-AC1-supplement.zip

---

## Author Comment (AC2) · 11 Feb 2019

We thank the reviewer for their time and their constructive and helpful comments. Our point-by-point responses are posted below, with the reviewer's comments being quoted first in italics.

*Comment:*

*1. A major portion of the findings (high photolability of tDOM; large tDOM contribution to the shelf DOM pool) echo findings in the companion paper (Martin et al. 2018). I suggest reframing the introduction and adding a paragraph briefly summarizing the findings of the companion manuscript and describing how the present study will build on this work. In particular, what you can learn from EEMs that hasn't been revealed with bulk DOC and CDOM analysis.*

Response:

We have summarized the findings from the companion paper in section 3.1 but we agree with the reviewer that it will be good as well to briefly summarize these findings at the end of the introduction as well. This will provide the readers with more background knowledge of the biogeochemical settings of dissolved organic carbon and colored dissolved organic matter. We will then explain how our study builds on the companion paper.

*Comment:*

*2. The calculation of tDOM appears to be oversimplified.*

*-Why is the sample with the highest value of normalized C1 Fmax used for the river endmember? Since there appears to be a lot of variation at 0 salinity, wouldn't an average be more appropriate? Is it possible to do a formal sensitivity analysis based on different choices of endmember?*

Response:

2.1 Selecting appropriate endmember values is of course an important aspect for our calculation. In our method, we used C1 as a quantitative tracer of terrigenous organic carbon, and selecting the highest C1/DOC value in the low salinity range as the terrestrial endmember makes our estimates more conservative. If we over-estimate the C1/DOC ratio of the freshwater end-member, our approach will correspondingly under-estimate the %tDOC in marine water, and therefore, we did not use the mean value of C1/DOC of all the freshwater samples. Using an average of the freshwater C1/DOC values would increase the final estimated %tDOC for the marine samples, by ~2 percentage point for March (since there is only tiny variability for the freshwater in March), and by up to 4 percentage points in September. The estimated range of %tDOC in September would then increase from 19%–45% to 20%–49%. Only one freshwater sample was collected in June. The difference between using the highest C1/DOC ratio or the mean value to do the calculation is not so pronounced for this estimation. We will include a brief summary of this information in the revised manuscript. Given that our resulting estimates of %tDOC firstly span a relatively large range, and also need to be interpreted cautiously, adding further sensitivity analysis is probably not warranted.

We realized that in our original Figure 6, the data from June and the data from September were accidentally switched. The correct Figure 6 is as below, which will replace the wrong one in the revised manuscript. We apologize for the mistake.

[Figure]

Figure 1. The correct Figure 6 in the manuscript (Estimated %tDOC).

*Comment:*

*-Equation 3 does not include a marine endmember, which implies that (1) C1 Fmax varies linearly with tDOC, and (2) C1 would be 0 in a hypothetical pure marine DOM sample. Both assumptions should be stated and justified. It is also assumed that C1 has the same reactivity as bulk tDOM despite representing a small, compositionally distinct fraction.*

Response:

This was also requested by Reviewer 1, and we will explicitly state our assumptions in the manuscript and add some further discussion to support them. Because the objective of our calculation was to gain an approximate estimate of the degree of terrigenous input to our most marine stations (which are still relatively close to shore and thus unlikely to represent true marine end-member waters devoid of terrigenous input), we necessarily had to assume that C1 was purely terrigenous in origin.

We believe that our assumptions are reasonable for the estimate of %tDOC within our relatively small study region because of the following three reasons.

(1) The predominantly conservative behavior of DOC concentration along the salinity gradient indicates that the distribution of DOC is mostly controlled by the mixing of freshwater and seawater, so our data do not suggest strong biogeochemical processing of the bulk DOC pool.

(2) Our C1 is very similar to terrigenous humic-like components identified in many other studies (Stedmon et al., 2003; Stubbins et al., 2014; Yamashita et al., 2015). Although we fully agree that fluorescent DOM only accounts for a small fraction of the total DOM pool, it has already been shown elsewhere that FDOM components are appropriate proxies for both fluorescent and non-fluorescent terrigenous DOM in the coastal aquatic environment, with strong correlations noted between fluorescent DOM measurements (including PARAFAC analysis) and molecular-scale measurements by mass spectrometry (Wagner et al., 2015). This indicates that our assumption that C1 behaves in the same way as non-fluorescent terrigenous DOM fractions during the freshwater-seawater mixing is in principle plausible. A more likely source of error in our study might be the preferential loss relative to non-fluorescent DOM of C1 caused by photo-degradation, given the high photo-lability of C1 found in this study. Preferential loss of C1 would lead us to under-estimate %tDOC in our marine samples, although the exact degree of C1 photo-lability needs to be better constrained in future experiments with South-East Asian peatland samples. However, because our C1 showed predominantly conservative mixing behavior across our sample set, our data do not suggest that C1 was rapidly and preferentially removed within our study region.

(3) Other studies have found only very low concentrations of C1-like FDOM components in the open ocean environment. For example, Murphy et al. (2008) reported only ~0.006 R.U. of a terrestrial humic-like component in the tropical Atlantic, which is ten-times lower than the values we observed at our fully marine stations in Sarawak. Since, unfortunately, we do not have open-ocean samples as a pure marine endmember, we necessarily have to assume that our C1 is purely terrestrial in origin. While this assumption may lead us to

slightly over-estimate %tDOC, existing open-ocean data do not suggest that this is a large source of error in our estimate (and, in fact, it would be counter-acted by the impacts of photo-degradation on C1).

We will add some additional discussion along these lines to justify our assumptions where appropriate in the manuscript.

*Comment:*

*-The identification of endmembers in Table S1 doesn't match the description in the text. There are marine end-members identified (not used in Eq 3), and some of the river endmembers are presented as an average of multiple stations instead of the station with the highest C1 Fmax as indicated in the text.*

Response:

The endmember stations listed in Table S1 are the endmembers we used for the conservative mixing models of the spatial distribution of PARAFAC components and HIX (Fig. 2). To estimate %tDOC, we used the highest value of C1/DOC of all freshwater stations in the Rajang and in the Samunsam so as to be more conservative, and these were not the same stations. We will describe this better in the revised manuscript to avoid confusion.

*Comment:*

*-Calculation of the %tDOC should be included in the methods section and more information provided.*

Response:

We explained the method of calculation of %tDOC in the discussion section because this is a derivative calculation based on the PARAFAC analysis that followed from our results of conservative behavior of C1 and DOC. We will re-consider the placement of this description in the revised version, and we will check to ensure that all necessary information is included.

*Other minor comments*

*Comment:*

*Methods: Photodegradation experiments should be in separate subsection.*

Response:

We will describe the photodegradation experiment methods with more details in a separate section, as also requested by Reviewer 1

*Comment:*

*Page 9, lines 13-24: paragraph mostly repeats info found elsewhere in the paper.*

Response:

We think that this section is actually necessary, because we discuss further the possible source of C4. Our analysis suggests strongly that our C4 has a terrestrial source, although it is conventionally associated more commonly with organic matter of marine origin. This is an important point in the paper, and together with the FDOM data from the Haroun et al. paper that we cite later in this section, our results will help to guide future efforts in this region to trace terrestrial carbon inputs. While there is inevitably a small degree of repetition, in Section 3.2 we simply compared the spectral characteristics of C4 with previous studies, without discussing the question of sources of the components.

*Comment:*

*Line 22: moreover is not the correct word here.*

Response:

We will change to "In addition" instead.

*Comment:*

*Page 10, lines 5-6: this sentence is contradictory.*

Response:

This sentence will be rephrased to state that the primary source of C5 in this study does not appear to be terrestrial.

*Comment:*

*Fig 4: legend indicates colors indicate regions in panel z.. figure appears to only go to panel y.*

Response:

Fig 4. The panel (z) was removed from this figure before submission. We will remove "panel (z)" from the caption as well. We are grateful to the reviewer for pointing this out. We apologize for the mistake.

---

## Author Response (AR1)

[revised manuscript text omitted]

**Response to Associate Editor's Comment**

We are grateful to the associate editor for their time in reviewing our manuscript and providing additional constructive suggestions regarding the sodium azide blank. Our detailed response is below, following the quoted associate editor's comments.

*Comment:*

*The authors have engaged constructively with the reviewers' comments but further work is required to the Level of a Major Revision addressing the reviewers (especially Reviewer #2) concerns. There is a specific technical issue of the use of Sodium Azide as a preservative of reactive DOM in the filtered water samples. This is a general practice based on NASA and COLOURS protocols codified in the REVAMP protocols ( G.H. Tilstone et al. 2002) which were developed and tested for the "Regional Validation of MERIS Chlorophyll products in North Sea coastal waters" .*

*The Azide ion itself has absorption in the UV and visible range. Tilstone etal. (2002) attribute 10% of the total absorption at 440 nm to the azide. This is not a fixed ratio but is determined by the relative amounts of azide and DOM and the UV absorption characteristic of the DOM at shorter wave lengths. The relative magnitude of this potential artefact will change at shorter wave lengths (in UV). Zhou et al. have addressed this potential weakness in their PARAFAC analysis by using the appropriate blanks. However, as noted by Referee #2 the problem may arise in the specific context of the determination of the spectral slope and the SUVA. While acknowledging these problems the author's response has been to assert that they may be discounted relying in part on the more detailed spectral measurements on the azide blank solutions reported in the supplement to the associated paper by Martin et al. The spectrum of sodium azide in water has been reported in the literature (McDonald et al. J. Chem.Phys. 52(1): 1332-1340 (1970)). However, their rebuttal lacks any quantitative detail. The authors are reluctant to pursue this issue further as it introduces a complicated technical /methodological discussion which is only marginally related to their main conclusions about the distribution of terrigenous DOM.*

*Reviewer #2 also notes the possible role of heavy metals complexed with the DOM interacting with the azide ion to produce coloured species which change the absorption spectra of the solutions. These effects will not be compensated for by the use of azide blanks.*

*These technical issues somewhat weaken, but do not vitiate, the main conclusions. The problem ultimately arises from the use of a protocol under circumstances which are outside its demonstrated range of validity. These problems are exacerbated by the logistic constraints of working on small boats in remote locations where the alterative sample preservation protocol by refrigeration are not to hand. My suggested resolution of this problem is to publish the paper subject to a major revision as outlined in the Authors response to the review comments but include an additional supplement where there is quantitative discussion of the likely size of perturbations due to the azide blank including the relative magnitude of the errors which arise and a demonstration that these effects may be accounted for and do not alter the conclusions.*

Response:

The suggestion to conduct a thorough analysis of the effects of $NaN_3$ blanks on CDOM parameters in a supplement is an excellent idea. As the editor says, our initial reluctance to go into this matter in detail in this manuscript was because it is somewhat peripheral to our analysis of FDOM, but as a supplement it does not distract from the main topics of this paper.

We have therefore written a detailed supplementary text that is referred to in our Methods section (Page 4), in which we quantitatively estimate the uncertainty that the $NaN_3$ blank introduces in the $SUVA_{254}$ and $S_{275-295}$. This analysis shows that the $NaN_3$ blank introduces only relatively small uncertainties in both parameters: the estimated total uncertainty in $SUVA_{254}$ is <10% for all samples (partly from $NaN_3$ blank and partly from DOC analytical error), and the estimated uncertainty in $S_{275-295}$ from the $NaN_3$ blank is <1% for all samples.

The reason why the $NaN_3$ blank introduces only little uncertainty in these parameters is because the concentration of $NaN_3$ was very consistent between all our samples. Thus, although the $NaN_3$ contributed a high proportion of the absorbance at short UV wavelengths, this blank could be quantified and subtracted with high accuracy, as we show in the Supplementary Information 1. We are grateful to the editor for suggesting this addition, which will hopefully alleviate any concerns that any readers may have. This said, we believe that the associate editor is mistaken in saying that "*The azide ion itself has absorption in the UV and visible range*". The absorption spectrum of sodium azide is presented in McDonald et al. (1970)that the editor referenced, and was also measured by us in the many $NaN_3$ blanks we prepared, and sodium azide in aqueous solution does not absorb at wavelengths above about 300 nm. The recommendation in the Tilstone et al. REVAMP protocols for using sodium azide is referenced back to a paper by Ferrari et al. (1996), in which the authors also state that sodium azide does not absorb above 300 nm. In the supplementary figures to the REVAMP protocols, Tilstone et al. (2002) show higher absorbance at all wavelengths from 300 – 800 nm for the spiked replicates compared to the non-spiked replicates. However, this is not consistent with the absorption spectrum of $NaN_3$, and is more suggestive of changes in the baseline than genuine blank absorption from a specific molecule. Our $NaN_3$ blanks do not show absorption above about 300 nm, as we discussed above, which is why we focused our uncertainty analysis just on CDOM parameters that are measured at wavelengths where $NaN_3$ genuinely absorbs light.

We agree with the reviewer and the editor that the possibility of Fe(III) interference for $SUVA_{254}$ needed addressing, and have added some additional discussion on this issue in the revised manuscript (P10 L10 – 14 in the revised version) . Unfortunately, we do not have Fe(III) measurements, so we cannot quantify the impact on $SUVA_{254}$. However, almost all of our $SUVA_{254}$ values are actually within a reasonable range (<5.5) while having very high decadic absorption coefficients, so we believe that Fe(III) is probably not a very significant factor in our samples. However, we agree that it is important to explicitly acknowledge this as a potential source of uncertainty. We note that the reviewer does not actually refer to any interactions between Fe(III) and $NaN_3$, and we are not aware of specific absorbent species that would be produced through such hypothetical reactions. We have therefore restricted ourselves to just addressing the reviewer's comments that separately raise the possibility of interference from Fe(III) and of interference from $NaN_3$.

References

McDonald, J. R., Rabalais, J. W. and McGlynn, S. P.: Electronic Spectra of the Azide Ion, Hydrazoic Acid, and Azido Molecules, J. Chem. Phys., 52(3), 1332–1340, doi:10.1063/1.1673134, 1970.

Tilstone, G. H., Moore, G. F., Doerffer, R., Røttgers, R., Ruddick, K. G., Pasterkamp, R. and Jørgensen, P. V: REVAMP Protocols Regional Validation of MERIS Chlorophyll products in, Work. Meet. MERIS AATSR Calibration Geophys. Valid. (ENVISAT MAVT-2003), (October), 1–77, 2002.

**Response to Reviewer 1**

We are very grateful for the reviewer's time and efforts spent on these helpful and constructive comments. Our responses to the reviewer comments are posted below, with the reviewer's comments quoted first in italics. We believe that we can address all of the reviewer's comments, and will revise our manuscript accordingly.

**2. Comments to Author**

*Note: I also read the companion paper 'Distribution and cycling of terrigenous dissolved organic carbon in peatland-draining rivers and coastal waters of Sarawak, Borneo' and the relevant review comments and the authors' answers to the comments.*

*This study (bg-2018-508) aimed to distinguish different fractions of dissolved organic matter (DOM) in peat-draining rivers, estuaries, and coastal waters of Sarawak, Borneo, using fluorescence spectroscopy and parallel factor (PARAFAC) analysis. The authors observed that the terrigenous fractions showed high concentrations at freshwater stations within the rivers, and conservative mixing with seawater across the estuaries, while the autochthonous DOM fraction showed low concentrations at all salinities. The authors claim that, based on the fluorescence data and little changes in optical properties of DOM, at least 20%–25% of the DOC at even the most marine stations (salinity >31) was terrestrial in origin. Although not all of the data provided is new to the relevant field, the content of this paper fulfills the requirements for the submission to Biogeosciences of which aims and scopes are to publish studies on all aspects of the interactions between the biological, chemical, and physical processes in terrestrial or extraterrestrial life with the geosphere, hydrosphere, and atmosphere. The title is representative of the article contents and the abstract summarize the contents clearly. Therefore, I recommend accepting this paper after the authors revise all the necessary points.*

*I have serious concerns about the use of sodium azide (NaN3) as a preservative for samples analyzed by UV absorption and fluorescence spectroscopy. Also, the emission wavelengths used to calculate fluorescence index (FI) seem inadequate. Finally, estimation of %tDOM by fluorescence is questionable.*

Response:

We are glad that the reviewer appreciates our study. We have addressed all of the specific concerns raised by the reviewer in our responses below.

*Comment 2.1.1 Estimate of terrestrial contribution*

*FDOM is only a small portion of the bulk DOM, and thus estimation of %tDOM by fluorescence is troublesome. PARAFAC components can be used to better understand biogeochemical processes that occur during the estuarine mixing, but PARAFAC components alone are not sufficient to estimate the tDOM contribution at given salinity. To make it possible, you must assume that all the rest of components in riverine DOM other than FDOM (PARAFAC component C1 in this case)*

*behaves in the same way as C1 does during the estuarine mixing and that marine end-member has no C1. Please explicitly state your assumptions. It's not enough in the current form.*

Response:

The reviewer correctly identifies the assumptions that underlie our estimate of %tDOC, *i.e.* that all the terrigenous DOM

fractions, both fluorescent and non-fluorescent, behave in the same way during the river-coastal sea interactions as C1, and that C1 represents terrestrial humic-like fractions that only come from terrestrial sources, while the marine environment in the open ocean has no C1. We agree that these assumptions need to be made clearer than in our original submission, and we have explicitly stated all these assumptions in the revised manuscript (bottom of P13 in the marked-up version).

We believe that our assumptions are reasonable for the estimate of %tDOC within our relatively small study region because of the following three reasons.

(1) The predominantly conservative behavior of DOC concentration along the salinity gradient indicates that the distribution of DOC is mostly controlled by the mixing of freshwater and seawater, so our data do not suggest strong biogeochemical processing of the bulk DOC pool.

(2) Our C1 is very similar to terrigenous humic-like components identified in many other studies (Stedmon et al., 2003;

Stubbins et al., 2014; Yamashita et al., 2015). Although we fully agree that fluorescent DOM only accounts for a small fraction of the total DOM pool, it has already been shown elsewhere that FDOM components are appropriate proxies for both fluorescent and non-fluorescent terrigenous DOM in the coastal aquatic environment, with strong correlations noted between fluorescent DOM measurements (including PARAFAC analysis) and molecular-scale measurements by mass spectrometry (Wagner et al., 2015). This indicates that our assumption that C1 behaves in the same way as non-fluorescent terrigenous DOM

fractions during the freshwater-seawater mixing is in principle plausible. A more likely source of error in our study might be the preferential loss relative to non-fluorescent DOM of C1 caused by photo-degradation, given the high photo-lability of C1 found in this study. Preferential loss of C1 would lead us to under-estimate %tDOC in our marine samples, although the exact degree of C1 photo-lability needs to be better constrained in future experiments with South-East Asian peatland samples. However, because our C1 showed predominantly conservative mixing behavior across our sample set, our data do not suggest that C1 was rapidly and preferentially removed within our study region. This is perhaps also because the spatial scales across which we sampled are ultimately not that extensive, so the tDOM residence time is probably still relatively short compared to the degradation rates of bulk tDOC and our C1.

(3) Other studies have found only very low concentrations of C1-like FDOM components in the open ocean environment. For example, Murphy et al. (2008) reported only ~0.006 R.U. of a terrestrial humic-like component in the tropical Atlantic, which is ten-times lower than the values we observed at our fully marine stations in Sarawak. Since, unfortunately, we do not have open-ocean samples as a pure marine endmember, we necessarily have to assume that our C1 is purely terrestrial in origin. While this assumption may lead us to slightly over-estimate %tDOC, existing open-ocean data do not suggest that this is a large source of error in our estimate (and, in fact, it might be counter-acted by the impacts of photo-degradation on C1).

We have added some additional discussion along these lines to justify our assumptions in the revised manuscript (bottom of P14 in the marked-up version).

*Comment:*

*Also, how do you explain %tDOC of >100% in Samunsam and Western Region (in March) at salinity >10 under your assumption?*

Response:

We agree that the few stations with %tDOC > 100% in the western region (mostly in March) are puzzling, and this
clearly calls for further work to investigate the use of FDOM as a quantitative tracer of tDOC. One possible reason is that the freshwater end-member value for C1/DOC ratio was underestimated for the Samunsam River. Because we could only collect a single freshwater sample in each season in the Western Region, the freshwater endmember might not be constrained sufficiently well. While the Samunsam does not have any large tributaries along the stretches we sampled, small channels from the surrounding mangroves do drain into the Samunsam estuary, so we cannot rule out additional inputs of C1-rich DOM at
mid-salinities. We note also that the Samunsam estuary is shallow, and especially in March there was a lot of resuspension of sediments at the mid-salinity stations that we sampled due to the strong tidal currents. Because terrestrial DOM can flocculate and/or be sorbed and desorbed from sediments, it is possible that resuspended sediments at these few estuarine stations acted as an additional source of C1. More FDOM and DOC data from this river system would ideally be needed to determine why the C1/DOC ratios at mid-salinities were higher than in the freshwater endmember. We have added some additional discussion
of this question to the appropriate part of the manuscript (P13 L26-P14L2 in the marked-up version).

*Comment:*

*In addition, Fmax/DOC is known to be susceptible to errors caused by the fluorescence intensity and DOC measurements (Korak et al. 2014), and the authors should include an evaluation of such an uncertainty (error propagation analysis),*
*since %tDOM estimation is I believe the most important part of this study.*

Response:

This is an important point concerning the accuracy of our estimate of %tDOC. As suggested by the reviewer, we have added an evaluation of uncertainty to the revised manuscript. For uncertainty analysis, we adopted ±4.3% uncertainty for DOC, based on the percentage uncertainty of repeated DOC measurements of the certified deep-sea water reference material (data from Martin et al., 2018). We adopted ±1% as the estimated error of the Fmax values of C1 (peak C) based on Korak et al., (2014). Formally propagating these uncertainties yields an uncertainty of around ±6% of the final tDOC estimate, so for a sample with 30% tDOC, the analytical uncertainty would amount to ±2% tDOC (so the sample would be estimated to have 30 ± 2% tDOC).

Because this analytical error is very small compared to the range of %tDOC that we estimate for our marine samples (which ranges by a factor of around 2), the analytical uncertainties are not really relevant. We have added a short explanation of this in the revised version (P13 L20-23 in the marked-up version).

*Comment:*

*Generally, in estuarine environments, contribution from estuarine vegetation (mangrove and marsh) is done by an end-member mixing model (0.1 salinity increment) using DOC concentrations of the fresh and marine end-member (Cawley et al. 2014). Because the main subject of this study is tropical peatlands, I feel that what the authors want to investigate is not riverine (derived from upper regions) inputs but inputs from the peatlands located in the estuary. The authors may reassess*

*contributions from the peatlands using the method reported in, for example, Cawley et al. (2014).*

Response:

We fully agree that a two-endmember mixing model using DOM concentrations of the freshwater and marine endmembers is an appropriate method for investigating DOM fluxes through estuaries to the sea. In coastal Sarawak, the companion paper (Martin et al., 2018) already conducted this analysis for DOC concentrations and in our study we use the same approach to study the distribution of FDOM components. We follow the same mixing model calculations as used by Cawley et al. (2014). Both in Martin et al. (2018) and the present study, we identified a conservative mixing pattern in the Western Region and additional input from the peatlands located in the delta of the Eastern Region, based on this mixing model approach. In this study, because we were able to decompose the FDOM as a mixture into multiple components representing different organic matter fractions from different sources, we could more confidently identify peatlands as the source of the additional DOC input along the Rajang Delta (as opposed to autochthonous production). However, the Rajang is the only one of our rivers in which the peatlands are exclusively located within the estuary, leading to the slightly non-conservative mixing pattern within the estuary. However, this could only be diagnosed because we used a mixing model based on a fully freshwater end-member station. Because we are already calculating mixing models as in Cawley et al. (2014) for all our rivers, we do not propose to make changes to these calculations.

*Comment 2.1.2 NaN$_3$*

*Although you said 'NaN3 did not contribute any blank fluorescence', it did contribute to sample absorbance, as you mentioned in the companion paper. Indeed sample preservation is still a major challenge, and I do use NaN3 to preserve samples for DOC analysis. However, I never use NaN3 to preserve samples for optical analysis because of the strong UV absorbance by NaN3 even at a low concentration (0.005% (w/v) in this study). I agree that if your samples have high absorbance, you could*

*correct for the NaN3 absorbance accurately. However, when measuring EEM for samples containing NaN3, it seems that you failed to correct for the inner-filter effects (IFEs) caused by NaN3, because for the IFEs correction you used the absorbance of CDOM that were obtained by subtracting the absorbance of NaN3 from that of samples containing NaN3. In that way, you underestimated fluorescence in the EEM regions where NaN3 absorbed light (Ex 250–280). This is very serious because you mentioned the protein-like component 'showed consistently low values across the study region', and this could be due to*

*underestimation of the protein-like component. The relative degree of the underestimation will be larger with decreasing sample absorbance relative to that of NaN3.*

*If you will correct (or may have corrected) for IFEs including NaN3 absorbance, please explain the degree of uncertainty of the correction. Because, although you said all samples had the same NaN3 concentration, there should be some variation in the concentration caused by, for example, repetitive volumetric measurements of samples (30 mL) and NaN3 solution (150*

*μL).*

Response:

The reviewer points out a critical aspect of EEM correction. We clearly did not explain the details of the inner filter effect corrections well enough. Indeed, we used the total absorbance of each sample (*i.e.*, absorbance of CDOM and $NaN_3$) for the inner filter effect correction. We then converted the fluorescence intensity to Raman Units, and then subtracted the fluorescence of our reagent blanks (DI water + $NaN_3$). Therefore, we do not underestimate the fluorescence intensity. This inner filter effect correction does not contribute any additional uncertainty from the presence of $NaN_3$, because the total absorbance of each individual sample was measured (we collected one single sample to measure both absorbance and fluorescence). Any variation in $NaN_3$ concentration between samples is therefore fully accounted for and included in the corrections. We have explained this important aspect more thoroughly in the methods section of the revised manuscript (P5 L8 in the marked-up version).

The issue of sample preservation with $NaN_3$ was already addressed in the discussion of the paper by Martin et al. (2018) in this issue: the reason we decided to try to use $NaN_3$ as a preservative was so we could follow the CDOM sampling protocols in use by the ocean remote sensing community, since our CDOM data are being used for remote sensing algorithm development. Given the problem with high blanks in the UV range, we would agree that this is not ideal for measurements below about 280

nm wavelength, but in our sample set we are very confident that we could correct for this blank with high accuracy, as discussed in Martin et al. (2018) and the accompanying discussion.

As also requested by the associate editor, we have added a full quantitative analysis of the uncertainty that the $NaN_3$ blank contributes to our estimates of CDOM parameters as Supplementary Information 1, where we analyzed the percentage contribution of $NaN_3$ blank to the total absorption coefficient, the uncertainty in $NaN_3$ blank absorption coefficients, uncertainty in SUVA$_{254}$ and uncertainty in S$_{275-295}$. To summarize this analysis, we find that, although the NaN$_3$ blank contributed a significant proportion of total sample absorption at short wavelengths, the uncertainty in the NaN$_3$ concentration in each sample was sufficiently small that the blank subtraction actually adds only a small amount of uncertainty to the estimated CDOM parameters.

*Comment 2.1.3 FI*

*Did you apply instrument-specific correction for EEM? If so, the emission wavelength for FI must be 470/520 nm instead of 450/500 nm (Cory et al. 2010; Kida et al. 2018), because the emission peak often lies between 450 and 500 nm when the correction applied, which makes FI meaningless (FI must be calculated on the right side of the emission peak). If not, please*
*write so in M&M section, because in that case your results are not directly comparable with other studies. It is often observed that if not corrected for the instrument-specific bias, the variability of FI between instruments is large for a given sample.*

Response:

We are grateful to the reviewer for pointing out this issue. We did indeed apply an instrument-specific correction, so
we have now re-calculated the FI using fluorescence intensities at 470/520 nm following Cory et al. (2010). This results in higher FI values for all samples, but the same distribution pattern along the salinity gradient, and the re-calculated FI values still show clear terrigenous signals for the blackwater rivers. The Eastern Region exhibits more mixed signals of terrestrial and microbial fulvic acids, but more towards the terrestrial endmember. We have interpreted the new FI results accordingly in the revised manuscript (P6 L24–29, P9 L6–27 in the marked-up version). However, we also note that the ranges in FI of terrestrial
*versus* microbial DOM endmembers are reported as quite variable in the literature, and the appropriate wavelength range to use for FI calculations is also still debated: even the paper mentioned by the reviewer (Kida et al. 2018) ultimately decided to calculate the FI at the traditional wavelengths of 450/500 nm because they judged the values at the longer wavelengths to be unreasonably high. We have included some extra discussion of this point in the revised manuscript (P9 L24–27 in the marked-up version).

*2.2 Minor comments*

*Comment:*

*Table 2. Was the distribution of the PARAFAC components and chlorophyll-a normally distributed? If not, Spearman's rank correlation should be used instead. Note that strong parametric linear relationships between PARAFAC components are*
*unlikely considering the theory of PARAFAC. If components have a strong linear correlation, PARAFAC cannot resolve these components and they appear as a single combined component. Correlations between PARAFAC components are generally expressed by a log-log plot or Fmax/DOC plot (Stedmon and Markager 2005).*

Response:

The PARAFAC components and chlorophyll-*a* were not normally distributed, so we have changed our correlation analyses to use Spearman's rank correlation (Table 2 in the marked-up version). Because the point of our correlation analyses is to show how our PARAFAC components co-vary with each other and with chlorophyll-a concentration across the salinity gradient, we decided not to normalize FDOM to DOC, because that would cancel out much of the variation that we are trying to analyze in this case. However, we agree of course with the point that the reviewer makes that very strong parametric linear relationships between PARAFAC components are ruled out by virtue of how PARAFAC models are calculated. However, this does not mean that PARAFAC components cannot be correlated with each other at all.

*Comment:*

*P2L26 'extremely high DOC concentrations' Please specify the DOC range, as it depends on person when a DOC concentration is 'extremely' high.*

Response:

The DOC range in the blackwater rivers in Sumatra and Borneo is up to 3000–5500 µmol L$^{-1}$ or 36–66 mg C L$^{-1}$, which lie on the highest DOC concentrations in the rivers reported globally. We have specified these numbers in the revised version (P2 L27 in the marked-up version).

*Comment:*

*P3L11 Sampling*

*How was the weather on the sampling days? In addition to seasonal changes, daily changes in rainfall and water flow conditions would affect DOM concentrations and compositions. If you discuss seasonal changes, at least the weather should be the same.*

Response:

During the sampling cruises, we did not encounter extreme weather events. Overall, during each expedition most days had part cloudy / part sunny weather conditions, and heavy rain showers of a few hours' duration occurred on many days (usually in the afternoon), as is typical for this equatorial climate. Cloud-free days were rare. Because a lot of the rainfall in this region takes place across small spatial scales, the weather conditions during any one day at any one particular location are not necessarily indicative of the weather across an entire river catchment. Hence, it is unlikely that DOM concentrations and composition were affected in a significant way by day-to-day changes in weather conditions, and indeed we do not see any evidence of this in our dataset. We have included some additional description to this effect in the methods section of the revised manuscript (P3 L21–24 in the marked-up version).

*Comment:*

*P3L30 Was the condition of the photodegradation experiment sterile (biodegradation-free)? If not, how about the effect of biodegradation? Please add some more details about the photodegradation experiment. For example, water inside the bottles was repetitively sub-sampled or you prepared many bottles and each bottle was collected as a sub-sample?*

Response:

The photo-degradation samples were filtered by 0.2-µm pore-size Anodisc filters to remove bacteria in order to rule out any effect of biodegradation or solubilization of particulate organic matter. Bottles were repetitively sub-sampled, and while this may have introduced some microbial contamination, this would have affected the dark control bottles to an equal extent. We have added these experimental details in the revised manuscript (P4 L11 – 13 in the marked-up version).

*Comment:*

*P4L3 'To minimize self-quenching of fluorescence intensity' Please add information on the maximum absorbance value of the measured samples, since IFE correction becomes invalid if sample absorbance is too high. Also, how you measured absorbance data is completely lacking. Please explain it in this section, and reference to the companion paper alone is not sufficient.*

Response:

We have included a brief summary in the M&M section revised manuscript of how the absorbance measurements were conducted (P4 L17 – 25 in the marked-up version): we used a dual-beam Thermo Evolution 300 spectrophotometer with quartz cuvettes, and selected a cuvette pathlength of either 100 mm, 10 mm, or 2 mm according to the sample absorbance (for the March data, high-absorbance samples were diluted with DI water because the 2-mm cuvette was unavailable).

For fluorescence measurements, we used a 1-cm cuvette for samples with low absorbance, while samples with high absorbance were either diluted 10-fold with DI water and then measured in a 1-cm cuvette (March samples), or measured undiluted in a 3-mm cuvette. For all samples, we used the $A_{total}$ of the appropriate dilution and pathlength at which the fluorescence measurements were conducted.

Kothawala et al. (2013) proposed that the inner filter effect correction is invalid for EEM regions with $A_{total} > 1.5$ because of non-linearity between absorbance and fluorescence intensity. We have three samples for which $A_{total} > 1.5$ in part of the EEM spectrum, as shown in Figure 1 below. Therefore, the PARAFAC results of these three samples, especially the C5, should be treated with caution. The $A_{total}$ values of all other samples are below 1.5 across the whole EEM, so the inner filter effect correction is fully valid for them. This information has been added to the M&M section (P5 L13 – 15 in the marked-up version) and the Figure 1 below has been added to the Supplementary Information as Figure S3.

[Figure]

Figure 1. Samples with Atotal above 1.5. The yellow shades indicate the regions where Atot > 1.5 in the respective EEMs.

*Comment:*

*P4L27 'chemical compound classes' The authors need to be careful here. What PARAFAC can do is to statistically deconvolute EEMs into underlying building blocks, termed 'components', and these components are rarely related to specific chemical compounds. I think the authors understood that, but for those who are not familiar with PARAFAC, the author's statement may be misleading.*

Response:

We fully agree with the reviewer that caution is needed here, hence we referred to "compound classes", not "compounds". We have re-phrase this as "…which decomposes the variation between EEMs in a dataset into multiple mathematically independent components representing different organic compound classes, with different sources, biogeochemical properties and behaviors." (P5 L27 in the marked-up version) to make this clearer also to non-specialists and avoid misunderstanding each PARAFAC component as a specific chemical molecule.

*Comment: P4L28 Specify how many samples were removed.*

Response:

Four samples were removed. We have added this information to the revised manuscript. (P5 L29 in the marked-up version)

*Comment:*

*P4L29 Please add in Fig. 3 the excitation and emission loadings of the validated split dataset.*

Response:

The excitation and emission loadings of the validated split dataset were saved during PARAFAC analysis and has been shown as an additional supplementary figure (Figure S4). It can provide further information about the validity of our five-component PARAFAC model for the readers.

*Comment:*

*P5L1 Fmax is not just a score value. "Fmax is calculated by multiplying the maximum excitation loading and maximum emission loading for each component by its score, producing intensities in the same measurement scale as the original EEMs"*

*(Murphy et al. 2013).*

*Also, Fmax cannot be a major of the concentration of each component in a sample, "because different fluorophores can have very different efficiencies at absorbing and converting incident radiation to fluorescence (Murphy et al. 2013)." Rather, "Quantitative and qualitative information may however be obtained from changes in the intensity of a given component, or in the ratios of any two components, between samples in the dataset (Murphy et al. 2013)."*

Response:

We have corrected our explanation of Fmax.

We fully agree that Fmax cannot indicate the absolute concentration of compounds, but instead indicates relative changes in concentration of each component between samples, which is the way we interpret our PARAFAC results throughout the manuscript. We realise that our description here was perhaps slightly misleading, so we have re-phrased the section to read "which is taken as a measure of the relative concentration of each component in different samples of a dataset" (P6 L2 in the marked-up version).

*Comment:*

*P5L8&L15 a350, S275–295, SR, and SUVA254 appeared for the first time here without explanations what they are. This is not kind for those who are not familiar with the optical indices. This is relevant to my comment on P4L3. Now I think that you need to make another section in M&M that explains the absorbance measurement and absorbance-based indices. However, personally I think that you can completely cut the sentences with respect to SR, a350 and SUVA254 since you mentioned about*

*SR and a350 only once or twice and did not discuss SUVA results (just correlation with HIX).*

Response:

We reviewed the need for mentioning each of these measurements, and have decided to omit all mention of the spectral slope ratio. We have added a brief explanation of any CDOM terms that are used (P4 L25 – 30 in the marked-up version). We believe that it is useful to briefly summarise these CDOM results from the companion paper in order to provide the readers with additional background about the CDOM concentration, DOM molecular weight and source in these rivers so that readers can appreciate the FDOM analysis more easily.

*Comment:*

*As for S275–295, you may want to use it to support your idea that an FDOM-based estimate of tDOM is OK. However, I am not totally convinced that being correlated with S275–295 supports the correctness of your fluorescence-based %tDOM (P11L24), because estimations of %tDOM based on S275–295 is non-linear (Fichot and Benner 2012).*

Response:

We agree that a correlation between our %tDOC estimates and $S_{275\text{-}295}$ does not prove the correctness of our method of calculating the %tDOC, but we do believe that it adds additional support: as in Fichot & Benner (2012), we find that there is an exponential relationship between %tDOC and $S_{275\text{–}295}$, as shown in Figure 2 below (%tDOC = exp (α + β $S_{275\text{-}295}$), where α=1.48, β=-126.23 ). We have added the exponential regression model of the relationship between %tDOC and $S_{275\text{-}295}$ to the
revised manuscript (P14 L28 – 30 in the marked-up version) and replaced Figure S1(b) with Figure 2 below.

[Figure]

Figure 2. Relationship between estimate of %tDOC and $S_{275\text{-}295}$. The observation data is plotted using circles and the exponential regression model is presented by the blue solid line.

*Comment:*

*P5L16 "SUVA254, 3.08–6.89" SUVA value of 6.89 is too high. Even the highest aromaticity sample (Ar >40%) in Weishaar et al. (2003) had the SUVA value of 5.3, and the possible maximum SUVA value (~5) has been recently suggested from a molecular analysis (Kellerman et al. 2018). Iron(III) is most probably interfering with SUVA determination in your sample*
*dataset (Poulin et al., 2014). If the authors did not measure Fe(III) and also have no stored sample for Fe(III) measurement, please state in the manuscript that some of SUVA values in this study was overestimated by interferences from Fe(III) to an unknown degree. Note that, if Fe(III) contributes to SUVA254 to a similar degree for all the samples, SUVA254 and SUVA280 would still have a high correlation.*

*Another possibility is the interference by NaN3 even after the blank correction. This is possible when the sample CDOM*
*absorbance was low. Please add the information on the NaN3 absorbance contribution to sample absorbance at 254 nm.*

*According to Fig S1&2 of the companion paper, decadic absorption coefficient of the NaN3 solution was about 4 m−1, which was about 10% - 30% of that of Rajang, Sematan, and Lundu samples and 50%–200% of marine samples. These values are not trivial.*

Response:

Unfortunately we do not have Fe(III) measurements, so we cannot rule out that the $SUVA_{254}$ values were impacted by iron. However, we note that peat-draining blackwater rivers typically have very low dissolved mineral concentrations. Moreover, our river water samples, especially those with highest $SUVA_{254}$, often had decadic absorption coefficients greater than 100 $m^{-1}$, so based on the data shown in the Poulin et al. (2014) paper, a very high Fe(III) concentration would be needed to significantly bias our estimates. We have added some discussion about this possible issue in the revised manuscript (P10 L10 – 14 in the marked-up version). While the Kellerman et al. paper is a very interesting study, we note that the authors only very tentatively propose an upper boundary of around 5.5 for $SUVA_{254}$, given their limited sample set, while other recent studies still use $SUVA_{254}$ up to 6.0 (*e.g.*Massicotte et al. (2017)). All but one of our samples had $SUVA_{254}$ below 6.0, and most samples were below 5.5, so even if our highest SUVA values are impacted by the presence of iron, this is unlikely to have affected our dataset to a very serious degree. Given the very large environmental gradients we sampled across, we think it is rather unlikely that the Fe(III) concentration was proportional to CDOM $a$(254) across all of our samples, so we still suspect that the strong correlation between $SUVA_{254}$ and $SUVA_{280}$ supports the reliability of our $SUVA_{254}$ estimates.

The possibility that $NaN_3$ was responsible for the high $SUVA_{254}$ values was already ruled out in the paper by Martin et al. (2018), given the very strong and linear relationship between $SUVA_{254}$ and $SUVA_{280}$, because $NaN_3$ no longer has any significant absorbance at 280 nm (besides, while the $NaN_3$ absorbance at 254 nm was certainly high, the $NaN_3$ concentration was kept very consistent between samples and was thus corrected for accurately). We have now quantitatively analyzed the uncertainty in $NaN_3$ and its effect on the uncertainties in $SUVA_{254}$ and $S_{275-295}$ in the supplementary information 1, where we also show the proportional contribution of the $NaN_3$ blank to sample absorbance for all samples from 250–320 nm.

*Comment:*

*P6L3 Please add seasonal climatic information (dry? rainy?) after months so that readers can easily understand climatic conditions, not only in the M&M section.*

Response:

We have added this where appropriate. It is important to note that in this equatorial climate there are not very distinct wet and dry seasons, instead, there is quite high rainfall year-round, that increases further during the wettest time of the year. We have highlighted this more clearly in the Methods section (P3 L20 – 21)

*Comment:*

*P9L31 "correlating strongly with DOC-normalized amino acid yields" This is not a correct citation.*

*The correlation coefficient was r = 0.62 (Fig. 8b in Yamashita et al., 2015), at best moderate correlation.*

Response:

This is a valid point. This sentence has been rephrased as "Furthermore, Yamashita et al. (2015) found that the DOC normalized protein-like component Fmax value was indicative of the amino acid content in DOM and thus the bioavailability of DOM." (P11 L14 in the marked-up version)

**2.3 Technical corrections**

*Comment:*

*P2L5 & L7 '0.2-0.25 Pg C yr-1' and '40% - 50%' should be 0.2–0.25 Pg C yr−1 and 40–50% (or 40%–50%). Please check the usage for minus (−), hyphen (-), en dash (–), and em dash (——). I did not correct for the rest of the manuscript.*

Response:

We have checked and corrected the usage of these symbols.

*Comment:*

*In Fig 2&4, it would be better to set the x axis to the same scale (maximum salinity of 35) except for the Simunjan River results so that comparisons between rivers become easier and more straightforward.*

Response:

We have set the x axis to the same scale except the Simunjan river as suggested by the reviewer to make the figures easier for the readers.

*Comment:*

*The caption of Fig. 4 says 'while they distinguish samples from different regions in the panel (z)', but I can't find the panel (z).*

Response:

The panel (z) was removed from the manuscript before submission, but we forgot to correct the caption. We have deleted "panel (z)" from the caption. We apologize for the mistake.

*Comment:*

*In Table 1, pleased add Tucker congruence coefficient (TCC) values so that readers can evaluate how much the comparisons are quantitative. Add the relevant explanations in M&M section as well.*

Response:

The tucker congruence coefficients between our models and the models from the cited literatures were all above 0.95, which indicates strong correlations. The respective TCC values can be found in the OpenFluor report attached as the Supplementary Data Table 2. Specifically, both Coble et al. (1996) and McKnight et al. (2001) did not run PARAFAC analysis so no TCC can be provided for them. We cited these two papers because the peak positions and spectra of our components are close to theirs identified by the peak-picking technique and they have been widely acknowledged as the nomenclature of FDOM EEM peaks. We are trying to keep the table concise and highlight the most critical information of the possible source and biogeochemical properties of the compound classes represented by our PARAFAC components so we were considering not add the respective TCC values for each pair of the models but this OpenFluor report has been uploaded as part of the Supplementary Information 3 and we have added the relevant explanations of TCC in the Table 1.

*-Why is the sample with the highest value of normalized C1 Fmax used for the river endmember? Since there appears to be a lot of variation at 0 salinity, wouldn't an average be more appropriate? Is it possible to do a formal sensitivity analysis based on different choices of endmember?*

Response:

2.1 Selecting appropriate endmember values is of course an important aspect for our calculation. In our method, we used C1 as a quantitative tracer of terrigenous organic carbon, and selecting the highest C1/DOC value in the low salinity range as the terrestrial endmember makes our estimates more conservative. If we over-estimate the C1/DOC ratio of the freshwater end-member, our approach will correspondingly under-estimate the %tDOC in marine water. This is why we did not use the mean value of C1/DOC of all the freshwater samples. Using an average of the freshwater C1/DOC values would increase the final estimated %tDOC for the marine samples. We have now calculated how big this difference is, and found that our estimate increases by ~2 percentage points for March (since there is only tiny variability for the freshwater in March), and by up to 4 percentage points in September. The estimated range of %tDOC in September would then increase from 19%–45% to 20%–

49%. Only one freshwater sample was collected in June. The difference between using the highest C1/DOC ratio or the mean value to do the calculation is not so pronounced for this estimation, relative to the range of %tDOC for our marine stations. We have included a brief summary of this information in the revised manuscript (P13 L12 – 16 in the marked-up version).

We realized that in our original Figure 6, the data from June and the data from September were accidentally switched. The correct Figure 6 is as below, which has replaced the wrong one in the revised manuscript. We apologize for the mistake.

[Figure]

Figure 1. The correct Figure 6 in the manuscript (Estimated %tDOC).

*Comment:*

*-Equation 3 does not include a marine endmember, which implies that (1) C1 Fmax varies linearly with tDOC, and (2) C1 would be 0 in a hypothetical pure marine DOM sample. Both assumptions should be stated and justified. It is also assumed that C1 has the same reactivity as bulk tDOM despite representing a small, compositionally distinct fraction.*

Response:

A more explicit statement of these assumptions was also requested by Reviewer 1, and we have now stated the underlying assumptions clearly in the manuscript, and added some further discussion to support them (P14 L16–32 in the marked-up version). Basically, we believe that our assumptions are reasonable for the estimate of %tDOC within our relatively small study region because of the following three reasons:

(1) The predominantly conservative behavior of DOC concentration along the salinity gradient indicates that the distribution of DOC is mostly controlled by the mixing of freshwater and seawater, so our data do not suggest strong biogeochemical processing of the bulk DOC pool.

(2) Our C1 is very similar to terrigenous humic-like components identified in many other studies (Stedmon et al., 2003; Stubbins et al., 2014; Yamashita et al., 2015). Although we fully agree that fluorescent DOM only accounts for a small fraction of the total DOM pool, it has already been shown elsewhere that FDOM components are appropriate proxies for both fluorescent and non-fluorescent terrigenous DOM in the coastal aquatic environment, with strong correlations noted between fluorescent DOM measurements (including PARAFAC analysis) and molecular-scale measurements by mass spectrometry (Wagner et al., 2015). This indicates that our assumption that C1 behaves in the same way as non-fluorescent terrigenous DOM fractions during the freshwater-seawater mixing is in principle plausible. A more likely source of error in our study might be the preferential loss relative to non-fluorescent DOM of C1 caused by photo-degradation, given the high photo-lability of C1 found in this study. Preferential loss of C1 would lead us to under-estimate %tDOC in our marine samples, although the exact degree of C1 photo-lability needs to be better constrained in future experiments with South-East Asian peatland samples. However, because our C1 showed predominantly conservative mixing behavior across our sample set, our data do not suggest that C1 was rapidly and preferentially removed within our study region. Ultimately, the spatial scales over which we sampled are not so large, and we suspect that the residence time of tDOM across this area is probably not very long relative to the degradation time-scales of tDOC and C1.

(3) Other studies have found only very low concentrations of C1-like FDOM components in the open ocean environment. For example, Murphy et al. (2008) reported only ~0.006 R.U. of a terrestrial humic-like component in the tropical Atlantic, which is ten-times lower than the values we observed at our fully marine stations in Sarawak. Since, unfortunately, we do not have open-ocean samples as a pure marine endmember, we necessarily have to assume that our C1 is purely terrestrial in origin. While this assumption may lead us to slightly over-estimate %tDOC, existing open-ocean data do not suggest that this is a large source of error in our estimate (and, in fact, it would be counter-acted by the impacts of photo-degradation on C1).

We have added some additional discussion along these lines to justify our assumptions in the revised manuscript (P13

L2 – 6 and P14 L16 – 32 in the marked-up version).

*Comment:*

*-The identification of endmembers in Table S1 doesn't match the description in the text. There are marine end-members identified (not used in Eq 3), and some of the river endmembers are presented as an average of multiple stations instead of the*

*station with the highest C1 Fmax as indicated in the text.*

Response:

The endmember stations listed in Table S1 are the endmembers we used for the conservative mixing models of the spatial distribution of PARAFAC components and HIX (Fig. 2 of the manuscript). To estimate %tDOC, we used the highest value of

C1/DOC of all freshwater stations in the Rajang and in the Samunsam so as to be more conservative, and these were not the same stations. We have described this more clearly in the revised manuscript to avoid confusion (P13 L12 in the marked-up version and Supplementary Data Table 1).

*Comment:*

*-Calculation of the %tDOC should be included in the methods section and more information provided.*

Response:

We explained the method of calculation of %tDOC in the discussion section because this is a derivative calculation based on the PARAFAC analysis that followed from our results of conservative behavior of C1 and DOC. We re-considered the placement of this description in the revised version, and we finally decided to leave it in the discussion section. We have checked to ensure that all necessary information is included (P13 L1 – 24 in the marked-up version).

*Other minor comments*

*Comment:*

*Methods: Photodegradation experiments should be in separate subsection.*

Response:

We have describe the photodegradation experiment methods with more details in a separate section (P4 L10 in the marked-up version), as also requested by Reviewer 1

*Comment:*

*Page 9, lines 13-24: paragraph mostly repeats info found elsewhere in the paper.*

Response:

We think that this section is actually necessary, because we discuss further the possible source of C4. Our analysis suggests strongly that our C4 has a terrestrial source, although it is conventionally associated more commonly with organic matter of marine origin. This is an important point in the paper, and together with the FDOM data from the Haroun et al. paper that we cite later in this section, our results will help to guide future efforts in this region to trace terrestrial carbon inputs. While there is inevitably a small degree of repetition, in Section 3.2 we simply compared the spectral characteristics of C4 with previous studies, without discussing the question of sources of the components.

*Comment:*

*Line 22: moreover is not the correct word here.*

Response:

We changed to "In addition" instead.

*Comment:*

*Page 10, lines 5-6: this sentence is contradictory.*

Response:

This sentence has been rephrased to state that the primary source of C5 in this study does not appear to be terrestrial (P11 L23 in the marked-up version).

*Comment:*

*Fig 4: legend indicates colors indicate regions in panel z.. figure appears to only go to panel y.*

Response:

Fig 4. The panel (z) was removed from this figure before submission. We have removed "panel (z)" from the caption as well. We are grateful to the reviewer for pointing this out. We apologize for the mistake.

List of all relevant changes in the revised manuscript

The numbers of pages and lines below are referring to the marked-up version.

P2 L27 Typical DOC concentration in blackwater rivers in Southeast Asia added

P3 L5 – 10 Brief summary of major findings from the companion paper (Martin et al. 2018) and explanation of how this study developed based on the companion study added

P3 L20 – 21 Explanation of dry/wet season in the study region added

P3 L21 – 24 Explanation of the weather condition during sampling trips added

P4 L11 – 13 Details of photodegradation experiment added

Photodegradation experiment method written in a separate section

P4 L17 – 25 Details of absorbance measurement added

P4 L25 – 30 Explanation of CDOM terms used in this paper added

P5 L8 More thorough explanation of using total absorption coefficient to conduct inner filter effect correction for the EEMs added

P5 L13 – 15 Explanation of Atotal values of our samples added

P5 L27 Explanation of PARAFAC rephrased

P5 L29 Number of samples that were removed during PARAFAC analysis added

P6 L2 Explanation of Fmax rephrased

P6 L24 – 29 and P9 L6 – 27 Re-calculated FI and interpretation added

P9 L24 – 27 Additional discussion on the selection of wavelengths to calculate FI added.

P10 L10 – 14 Additional discussion on the potential interference of Fe(III) on $SUVA_{254}$

P11 L14 Citation of Yamashita et al. 2015 paper rephrased

P11 L24 Sentence rephrased

P13 L2 – 6 Further explanation of the assumptions of estimate of %tDOC added

P13 L12 – 16 Summary of the variability of %tDOC due to different ways of selecting riverine endmembers added

P13 L20 – 23 Explanation of uncertainty in the Fmax of C1/DOC added

P13 L26 – P14 L2 Discussion on the %tDOC > 100% in four stations in the Western Region added

P14 L16 – 32 Additional discussion on the assumptions of our estimate of %tDOC added

P14 L28 – 30 Regression model of the relationship between %tDOC and $S_{275-295}$ added

Table 2 Spearman's rank correlation used

Figure 2&4 the x axis reset

Figure 4 "panel (z)" in the caption removed

Figure 6 Panel b and c swapped

Supplementary Figure 1b (Figure S1b) replaced by the figure of exponential regression model of the relationship between %tDOC and $S_{275-295}$

Supplementary Figure 3 (Figure S3) added: Samples with Atotal > 1.5

Supplementary Figure 4 (Figure S4) added: excitation and emission loadings of the validated split dataset.

Supplementary Data Table 1 Riverine endmember of estimate of %tDOC added

Supplementary Data Table 2 added: Open-Fluor report with Tucker Congruence Coefficients.

Supplementary Information 1 added to explain the quantitative analysis of the uncertainty in $NaN_3$ blank and its effect on the CDOM parameters.

---

## Referee Report (RR1)

**Second Submission**

Ms. Ref. No.: bg-2018-508

Title: Composition and cycling of dissolved organic matter from tropical peatlands of coastal Sarawak, Borneo, revealed by fluorescence spectroscopy and PARAFAC analysis

Authors: Yongli Zhou, Patrick Martin, Moritz Müller

**1. Overview and general recommendation**
**1.1. Recommendation**
Accept after minor revision

No additional review is needed as long as all the comments below will be addressed.

**1. 2. Comments to Author**
Thank you for your revision. I am satisfied with the revision and have only minor comments.

Regarding your responses to the editor, I agree with you. $NaN_3$ does not absorb >300 nm and I did not actually refer to any interactions between Fe(III) and $NaN_3$.

**2. Minor comments**
2.1. Supplementary information about $NaN_3$ (relatively major comment)

I appreciate the authors' efforts. However, is the label for the Y axis in Figure S1 maybe wrong? The percentage contribution of $NaN_3$ blanks to the total Napierian absorption coefficient (CDOM + $NaN_3$) reaches almost 100%, which means that the absorbance of solution containing CDOM + $NaN_3$ was almost completely occupied by $NaN_3$. The highest value seems around 95%. This means that $NaN_3$ adsorbed 95% and CDOM adsorbed only 5% ($NaN_3$ 19 times higher than CDOM)? Perhaps the Y axis should be 'The percentage contribution of $NaN_3$ blanks to the CDOM Napierian absorption coefficient?' If so, 100% means the equal contribution from $NaN_3$ and CDOM. But you're indeed saying 'The $NaN_3$ accounts for 0 – 95% of the total absorption coefficient at 250 nm'….

2.2. PARAFAC should be written out at the first use. P3L8

2.3. Were water samples for the photodegradation experiment sterilely filtered directly into quartz bottles? It's unclear. P3L22 & P4L5

2.4. No year for the citation McDonald et al. P4L15

2.5. References to $S_{275-295}$ and $SUVA_{254}$ are needed. P4L18

2.6. Were the sample fluorescence intensities normalized before PARAFAC? P5L15

2.7. Were the removed four outliers during PARAFAC modeling projected onto the validated model later and their results are reported? Did the model fit well? P5L20

2.8. Interpretation of FI values according to Cory et al. (2010) P4L12–L18 & P8L21–L30 (relatively major comment)

After correction for all spectroscopic biases, and with Fluoromax, FI only varies from 1.2 to ~1.6 (Cory et al., 2010). traditional FI range of 1.2–1.9 (McKnight et al., 2001) is no longer valid. Your data showed that FI ranged between 1.1 and 1.6, which is a full range that FI can take. Thus, saying `The fluorescence index (FI) was very low across the whole study region` while citing Cory et al. (2010) does not make sense to me. I suggest you remove McKnight paper and change the interpretation of FI that reflects Cory et al. (2010). In fact, both papers were written by the same authors and thus they changed the interpretation, but still other researchers are citing the old paper. One possible cause of this is that the original authors changed their FI calculation in Cory and McKnight et al. (2005) paper WITHOUT saying any reason. Nine years later they revealed the reason for the first time. However, you can refer to the paper by Maie et al (2006) 'Chemical characteristics of dissolved organic nitrogen in an oligotrophic subtropical coastal ecosystem' (*2.3 Optical measurements*), where you can find that they knew the reason beforehand (they knew the Cory's Ph.D. dissertation).

2.9. FI as a tDOM tracer P9L9 & P14L5

Related to the previous comment. FI value of 1.5 or 1.6 does not ensure that your coastal sample is dominated by tDOM. Nevertheless, I totally agree with your assertion that FI is not a good indicator of DOM origin. It's likely that the fluorescence indices generally work better in the order of HIX>BIX>FI.

2.10. Assumption for tDOM estimate by PARAFAC

The sentences 'Our approach assumes firstly that C1 is exclusively terrestrially derived, and has no non-terrestrial sources in estuaries and marine waters (P12L28)….The first assumption is probably broadly valid: as discussed above, Fmax values of C1-like components in open-ocean waters are very low relative to the values across our study area (P12L30)'

and

'Alternatively, there could be additional sources of C1-rich DOM within the Samunsam estuary (P12L23)' and 'In the Rajang River, C1–C4 all showed positive deviations from conservative mixing,

suggesting that there were additional inputs of all of these components in the Rajang estuary (P7L24)' are counter-intuitive.

To me, it seems you are still struggling with the use of C1 as a tDOM tracer because of additional C1 inputs from estuarine environments. However, the estuarine vegetation (peat and/or mangrove) is also terrestrial. Maybe you can solve the problem by including this vegetation to tDOM sources?

fin

---

## Author Response (AR2)

[revised manuscript text omitted]

Response to Referee Report 1

We are grateful to the reviewer's time and efforts again for all these constructive comments and suggestions. Our point-by-point response is posted below, with the reviewer's comments quoted first in italics followed by our response. We have addressed all the concerns proposed by the reviewer and revised the manuscript accordingly.

*Comment:*
*2.1. Supplementary information about NaN3 (relatively major comment)*

*I appreciate the authors' efforts. However, is the label for the Y axis in Figure S1 maybe wrong? The percentage contribution of NaN3 blanks to the total Napierian absorption coefficient (CDOM + NaN3) reaches almost 100%, which means that the absorbance of solution containing CDOM + NaN3 was almost completely occupied by NaN3. The highest value seems around 95%. This means that NaN3 adsorbed 95% and CDOM adsorbed only 5% (NaN3 19 times higher than CDOM)? Perhaps the Y axis should be 'The percentage contribution of NaN3 blanks to the CDOM Napierian absorption coefficient?' If so, 100% means the equal contribution from NaN3 and CDOM. But you're indeed saying 'The NaN3 accounts for 0 – 95% of the total absorption coefficient at 250 nm'....*

Response:

Thanks the reviewer for posting this concern, which gives us a chance to further clarify our analysis of the interference of $NaN_3$ blank to the CDOM absorbance. In this analysis, we divided the Napierian absorption coefficients of the $NaN_3$ blank by the TOTAL Napierian absorption coefficients of each sample (CDOM+$NaN_3$) for each wavelength to obtain the percentage contribution of $NaN_3$ to the TOTAL sample (CDOM+$NaN_3$) at different wavelengths, thus the Y-axis label in the Figure 1 of the Supplementary Information 1 is correct.

As the reviewer mentioned, the percentage contribution of the $NaN_3$ blank to the total sample did reach up to around 95% (i.e. 95% of the total absorbance is from $NaN_3$ and 5% is from CDOM), but such high blank contribution was only observed at very short wavelengths (<260 nm) for a few samples from the marine stations with extremely low CDOM absorbance. Furthermore, the blank contribution rapidly decreases to below 15% beyond 270 nm and to 0% beyond 310 nm for most of the samples, so the interference from the $NaN_3$ to the CDOM analysis is limited.

As the Y-axis of Figure 1 of the Supplementary Information 1 mentioned by the reviewer is indeed correct, we have not made any change to it.

*Comment:*
 *2.2. PARAFAC should be written out at the first use. P3L8*

Response: PARAFAC is written out in P3L8 now (P3L8 in the marked-up

version).

*Comment:*

*2.3. Were water samples for the photodegradation experiment sterilely filtered directly into quartz bottles? It's unclear. P3L22 & P4L5*

Response: We have clarified this now (P4L5—L9). The water samples for the photodegradation experiment were filtered and then combined first in a freshly acid-washed 1-L glass bottle, and then poured directly into the quartz bottles. This ensured that all bottles started off with identical water. While 0.2 µm filtration should yield a sterile water sample, we were unable to guarantee a perfectly sterile working environment on the boat, so we have not claimed that the filtration was sterile – any effect of bacterial contamination was accounted for by the dark controls. The section on P3L22 only pertains to the main water samples, not the samples for the photodegradation experiment, so we have not made changes on P3L22.

*Comment:*

*2.4. No year for the citation McDonald et al. P4L15*

Response:

Thanks the reviewer for pointing this mistake out. The year for the citation has

been added (P4L18 in the marked-up version).

*Comment:*

*2.5. References to S275–295 and SUVA254 are needed. P4L18*

Response:

The references have been added (P4L23 – L24 in the marked-up version but the changes seem to be untracked when we use Mendeley to add the references).

*Comment:*

*2.6. Were the sample fluorescence intensities normalized before PARAFAC? P5L15*

Response:

Yes the sample fluorescence intensities were normalized using the "normeem" function in the drEEM toolbox before PARAFAC to ensure that the intensities from different samples are comparable in order to avoid bias on the samples with high fluorescence intensity but the original (i.e. un-normalized) intensities were used to calculate the Fmax. Because normalization is a necessary step, which is not optional, when ones are using the drEEM toolbox to run PARAFAC analysis, we decided not to add this information to the manuscript.

*Comment:*

*2.7. Were the removed four outliers during PARAFAC modeling projected onto the validated model later and their results are reported? Did the model fit well? P5L20*

Response:

Sorry for not writing it clearly. No, they were not projected onto the validated model because we believe that the FDOM spectra for those four outliers are erroneous, probably due to contamination or false operation during measurement, so they were completely removed from our dataset. We have stated that the four outliers were removed "from this study" (P5L23 in the marked-up version).

*Comment:*

*2.8. Interpretation of FI values according to Cory et al. (2010) P4L12–L18 & P8L21–L30 (relatively major comment)*

*After correction for all spectroscopic biases, and with Fluoromax, FI only varies from 1.2 to ~1.6 (Cory et al., 2010). traditional FI range of 1.2–1.9 (McKnight et al., 2001) is no longer valid. Your data showed that FI ranged between 1.1 and 1.6, which is a full range that FI can take. Thus, saying `The fluorescence index (FI) was very low across the whole study region` while citing Cory et al. (2010) does not make sense to me. I suggest you remove McKnight paper and change the interpretation of FI that reflects Cory et al. (2010). In fact, both papers were*

*written by the same authors and thus they changed the interpretation, but still other researchers are citing the old paper. One possible cause of this is that the original authors changed their FI calculation in Cory and McKnight et al. (2005) paper WITHOUT saying any reason. Nine years later they revealed the reason for the first time. However, you can refer to the paper by Maie et al (2006) 'Chemical characteristics of dissolved organic nitrogen in an oligotrophic subtropical coastal ecosystem' (2.3 Optical measurements), where you can find that they knew the reason beforehand (they knew the Cory's Ph.D. dissertation).*

Response:

We really appreciate this comment as it is important for correctly interpreting the FI results. We apologize firstly for the misinterpretation that FI from coastal Sarawak ranging from 1.1 to 1.6 was "low". Cory et al. (2010) reported that the corrected-FI of microbially derived DOM and that of terrestrially derived DOM are typically ~1.5 and ~1.2, respectively, with 1.4 as the rough threshold that distinguishes the two DOM pools. Therefore, the FI values consistently below 1.4 for the mid- and low salinity range (S<20) across the entire study region, except the Rajang River, suggest that a large proportion in the total DOM pool is derived from land. The increasing FI values along the salinity gradient tracks the shifts in the composition of the total DOM pool, towards higher relative contribution from the microbially derived DOM in the marine environments. However, the FI values of ~1.5 for the entire Rajang River and eastern region do not necessarily

indicate the dominance of microbial DOM because multiple lines of evidence have been found to show significant terrestrial DOM signals ($S_{275-295}$, C1 – C4 Fmax, HIX, etc.) in the Rajang River.

We have re-interpreted the FI results accordingly (P6L17-L21, P9L4-L9, P9L12-L14, P9L18-L19, P14L24-L26 in the marked-up version).

*Comment:*

*2.9. FI as a tDOM tracer P9L9 & P14L5*

*Related to the previous comment. FI value of 1.5 or 1.6 does not ensure that your coastal sample is dominated by tDOM. Nevertheless, I totally agree with your assertion that FI is not a good indicator of DOM origin. It's likely that the fluorescence indices generally work better in the order of HIX>BIX>FI.*

Response:

We agree with the reviewer and have made revisions regarding the interpretation of the FI values of the coastal water samples (P6L17-L21, P9L4-L9, P9L12-L14, P9L18-L19, P14L24-L26 in the marked-up version ).

*Comment:*

*2.10. Assumption for tDOM estimate by*

*PARAFAC*

*The sentences 'Our approach assumes firstly that C1 is exclusively terrestrially*

*derived, and has no non-terrestrial sources in estuaries and marine waters (P12L28)….The first assumption is probably broadly valid: as discussed above, Fmax values of C1-like components in open-ocean waters are very low relative to the values across our study area (P12L30)'*

*and*

*'Alternatively, there could be additional sources of C1-rich DOM within the Samunsam estuary (P12L23)' and 'In the Rajang River, C1–C4 all showed positive deviations from conservative mixing, suggesting that there were additional inputs of all of these components in the Rajang estuary (P7L24)' are counterintuitive.*

*To me, it seems you are still struggling with the use of C1 as a tDOM tracer because of additional C1 inputs from estuarine environments. However, the estuarine vegetation (peat and/or mangrove) is also terrestrial. Maybe you can solve the problem by including this vegetation to tDOM sources?*

Response:

We agree with the reviewer that there are clearly complexities in using C1 as a tDOM tracer. In the Rajang, the peatlands that make up the river delta are most likely the additional source of C1—C4 for the mid-salinities of the Rajang River, which has been included in the revised manuscript (P14L4-L8 in the marked-up version). However, this might not be the case for the Samunsam River because the peatlands are only located at its upstream region. We agree with the reviewer

that mangrove vegetation could be an additional source of C1-rich DOM. In fact, we had tried to state this in the revised version already, and we have now made this more explicit by changing "mangroves" to "mangrove vegetation" (P13L9 in the marked-up version). We further suggest that the additional C1-enriched DOM compounds could be possibly desorbed from suspended sediments. Clearly, follow-up work on these questions would be needed.

References

Cory, R. M., Miller, M. P., Mcknight, D. M., Guerard, J. J. and Miller, P. L.: Effect of instrument-specific response on the analysis of fulvic acid fluorescence spectra, Limnol. Oceanogr. Methods, 8(2), 67–78, doi:10.4319/lom.2010.8.67, 2010.

List of all relevant changes made

P4L5—L9: Clarified the details of filtration of the photodegradation experiment.

P3L8: "PARAFAC is written out for the first time of use"

P4L23 – L24: References added.

P4L18: The year for the citation added.

P5L23: We clarified that the four outliers were removed "from this study".

P6L17-L21, P9L4-L9, P9L12-L14, P9L18-L19, P14L24-L26: re-interpretation of the FI results.

P13L9: "mangroves" changed to "mangrove vegetations"